# Nonhomologous tails direct heteroduplex rejection and mismatch correction during single-strand annealing in *Saccharomyces cerevisiae*

**Elena Sapède, Neal Sugawara, Randall G. Tyers, Yuko Nakajima, Mosammat Faria Afreen, Jesselin Romero Escobar, James E. Haber** [ID] *

Rosenstiel Basic Medical Sciences Research Center and Department of Biology, Brandeis University, Waltham, Massachusetts, United States of America

* haber@brandeis.edu

**Data Availability Statement:** Details and code are available from: https://github.com/users/RandallTyers/FA_variant_processor. DNA

## Abstract

Single-strand annealing (SSA) is initiated when a double strand break (DSB) occurs between two flanking repeated sequences, resulting in a deletion that leaves a single copy of the repeat. We studied budding yeast strains carrying two 200-bp *URA3* sequences separated by 2.6 kb of spacer DNA (phage lambda) in which a site-specific DSB can be created by HO or Cas9 endonucleases. Repeat-mediated deletion requires removal of long 3'-ended single-stranded tails (flaps) by Rad1-Rad10 with the assistance of Msh2-Msh3, Saw1 and Slx4. A natural 3% divergence of unequally spaced heterologies between these repeats (designated F and A) causes a significant reduction in the frequency of SSA repair. This decrease is caused by heteroduplex rejection in which mismatches (MMs) in the annealed intermediate are recognized by the MutS (Msh2 and Msh6) components of the MM repair (MMR) pathway coupled to unwinding of the duplex by the Sgs1-Rmi1-Top3 helicase. MutL homologs, Mlh1-Pms1 (MutL), are not required for rejection but play their expected role in mismatch correction. Remarkably, heteroduplex rejection is very low in strains where the divergent repeats were immediately adjacent (Tailless strains) and the DSB was induced by Cas9. These results suggest that the presence of nonhomologous tails strongly stimulates heteroduplex rejection in SSA. DNA sequencing analysis of SSA products from the FA Tailed strain showed a gradient of correction favoring the sequence opposite each 3' end of the annealed strand. Mismatches located in the center of the repair intermediate were corrected by Msh2-Msh6 mediated mismatch correction, while correction of MMs at the extremity of the SSA intermediate often appears to use a different mechanism, possibly by 3' nonhomologous tail removal that includes part of the homologous sequence. In contrast, in FA Tailless strains there was a uniform repair of the MMs across the repeat. A distinctive pattern of correction was found in the absence of *MSH2*, in both Tailed and Tailless strains, different from the spectrum seen in a *msh3Δ msh6Δ* double mutant. Previous work has shown that SSA is Rad51-independent but dependent on the strand annealing activity of Rad52. However Rad52 becomes dispensable in a Tailless

sequence files for Fig 7 and data for other figures are stored at https://zenodo.org/records/10498510.

**Funding:** National Institute of General Medical Sciences R35 GM127129.

**Competing interests:** The authors have declared that no competing interests exist.

construct where the DSB is induced by Cas9 or in transformation of a plasmid where SSA occurs in the absence of nonhomologous tails.

## Author summary

Deletions between dispersed, divergent, repeated sequences are frequently found among cancer mutations. We have studied deletion formation by single-strand annealing (SSA) between naturally divergent 200 bp repeats in budding yeast. SSA was initiated by creating a site-specific double-strand break (DSB) in the region between the two repeats, either using HO endonuclease or guide RNA-directed Cas9. In this process, the ends are resected 5' to 3', leaving long 3'-ended single-stranded sequences that can anneal. After annealing any 3'-ended nonhomologous tails are clipped off by the Rad1-Rad10 endonuclease. The presence of nonhomologous tails triggers a Msh2-Msh3-dependent and Sgs1-dependent heteroduplex rejection when there are 7 heterologies within the 200-bp repeats being annealed, whereas a "Tailless" construct is not rejected. These results suggest that the presence of nonhomologous tails strongly stimulates heteroduplex rejection in SSA. Mismatch repair within the heteroduplex is Msh2-Msh6-dependent; however, a *msh2* deletion proved have a spectrum of mismatch repair different from a *msh3 msh6* double mutant, implying that Msh2 may have some additional role when the intermediates have nonhomologous 3' tails. "Tailed" repeats yielded a gradient of repair that favored sequences in the "left" repeat, but "Tailless" SSA yielded a uniform correction of all 7 heterologies in the 200-bp region. A cluster of repeats near the left end of the sequence provokes correction of these markers in a Msh6-independent fashion; this correction is not attributable to 3' to 5' exonuclease activity of DNA polymerase δ and may reflect instances where part of the homologous sequence itself is clipped off by Rad1-Rad10. The presence or absence of nonhomologous tails also affected the requirement for the Rad52 strand annealing protein. Both in the chromosomal SSA assay and in a plasmid assay, Rad52 was largely dispensable when there were no nonhomologous tails after strand annealing.

## Introduction

In the human genome, repetitive DNA sequences such as LINEs (long interspersed nuclear elements) and SINEs (short interspersed nuclear elements, such as Alu) constitute at least 34% of the genome [1]. Recombination between repetitive DNA sequences can lead to chromosomal translocations, deletions, or inversions which are thought to contribute to tumor formation [2]. As one example, many of the mutations in *BRCA1*, a gene known to be linked to breast and ovarian cancers are due to deletions and duplications between Alu repeats [3]

One way in which a double-strand break (DSB) can be repaired is through single-strand annealing (SSA) between repeated sequences flanking the DSB, resulting in a chromosomal deletion retaining one copy of the repeated sequence, also referred to as repeat-mediated deletion (RMD). SSA requires extensive 5' to 3' resection of the DSB ends to expose complementary single-strand DNA (ssDNA) sequences that can anneal, producing an intermediate in which there are 3'-ended single-stranded nonhomologous tails. Both in yeast and in mammals, SSA can occur between repeats that are as far apart as 25 kb [4,5]. Extensive resection in yeast depends on either the Exo1 or Sgs1-Rmi1-Top3-Dna2 exonucleases; in mammals the Sgs1 homolog BLM appears to play a similarly important role in resection [6]. Strand annealing in

budding yeast requires the Rad52 annealing protein but is independent of the Rad51 recombinase that is required for most other types of homologous recombination [7,8]. In mammals, Rad52 appears to be one of several annealing proteins [6]. In yeast, completion of SSA requires the excision of nonhomologous tails by the Rad1-Rad10 (ERCC1-XPF) 3' flap endonuclease, Saw1 [9] and the Slx4 scaffold protein [10,11]. Efficient flap removal in yeast is also dependent on the MutSβ Msh2 and Msh3 mismatch repair proteins that recognize branched DNA structures [12,13].

The distance between the DSB and the repeated fragments defines the amount of end resection required for SSA repair. In *S. cerevisiae*, a DSB created between repeats as far apart as 25 kb led to the efficient deletion of the separating fragment [5]. Mutations such as *fun30Δ* and *rad50Δ* that impair resection prevent completion of SSA when the repeats are separated by a long distance [13]. In mammals, studies on mouse embryonic stem cells showed that end resection is critical for SSA at least within the first few kb (3.3 kb) [4]. Especially when resection is limited, repeat mediated deletions can also be carried out by Rad51-dependent break-induced replication (BIR) [14].

Besides the distance between the repeated fragments, another parameter that influences the efficiency of SSA is the degree of homology between the repeats. In both yeast and mammals, a low level of heterology—as little as 3%—decreased SSA repair more than two-fold [4,13]. In budding yeast, this reduction is caused by heteroduplex rejection, in which the annealed resected strands of the repeat are unwound by the Sgs1-Rmi1-Top3 helicase complex, apparently prompted by the binding of the MutSα Msh2-Msh6 mismatch repair proteins, but independent of the MutLα Mlh1-Pms1 mismatch repair proteins that are still required for mismatch correction [13,15]. MutL homolog proteins have been implicated in heteroduplex rejection in other systems; studies on strains containing homologous and divergent substrates highlighted the role of Mlh1 and Msh6 in distinguishing between crossover and non-crossover events [16]. Heteroduplex rejection does not involve the degradation of the annealing sequences, as they can participate in other, more distant SSA events with a fully homologous partner [13,17]

Several previous studies provided evidence that 3' non-homologous tail could be important for heteroduplex rejection [13,15]. Chakraborty et al. showed the rejection is favored before 3' non-homologous tail clipping during SSA, this being a critical regulatory step in the repair vs. rejection decision [13,15] However, the close relationship between heteroduplex rejection and the removal of a 3'-ended nonhomologous is not fully understood. In a Rad51-dependent recombination process such as break-induced replication (BIR), the presence of 1 or 2 mismatches between 108-bp regions undergoing recombination had little or no effect when the DSB end was perfectly aligned with the donor, but these same heterologies reduced BIR substantially when the DSB end carried a 68-nt nonhomologous tail [18]; even a 3-nt tail had a significant effect. Thus, we wished to assess how nonhomologous tails affected heteroduplex rejection in SSA.

We used a simple SSA assay [13] in which a site-specific DSB is induced between two 200-bp regions located in the 5' promoter region of the *URA3* gene on chromosome V. These duplicated regions were derived from two different *S. cerevisiae* strains, designated F and A, which harbor six single base substitutions and a T insertion in F relative to A (Fig 1A and S1 Table); these naturally occurring heterologies are nonrandomly spaced as shown in Fig 1C and S1 Table. The insertion/deletion of T is found in a run of 11/12 Ts immediately adjacent to mismatch 3 (MM3).

In the FA Tailed strain, the F and A repeats are separated by a 2.6 kb spacer composed of phage λ, pUC9 DNA and a 117-bp recognition site for the site-specific HO endonuclease [10,11] (Fig 1A) [13,17]. DSBs can also be created at different locations in the phage λ DNA

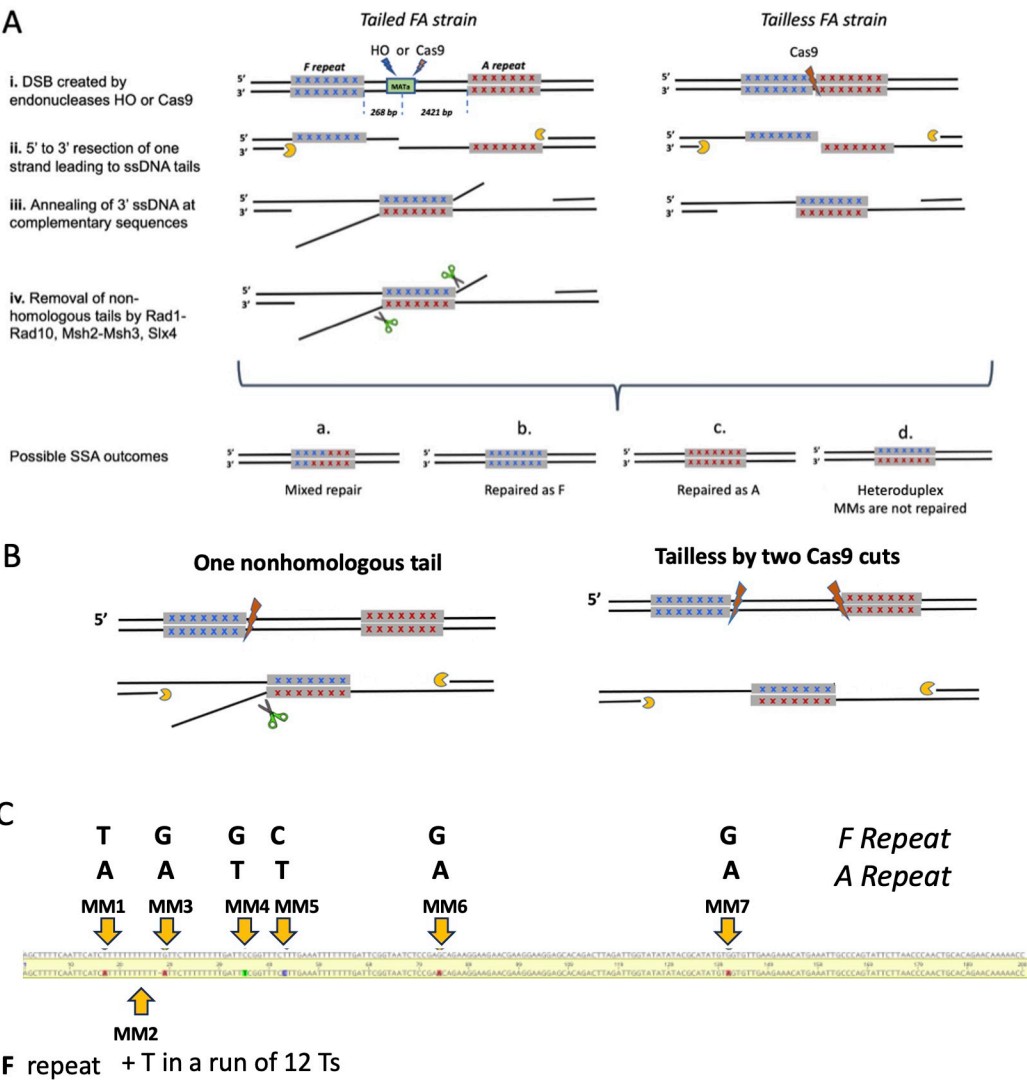

**Fig 1. SSA repair in *S. cerevisiae*.** SSA repair between divergent repeats. (i) A DSB is created by HO or Cas9 endonucleases between two 205-bp repeated sequences in Tailed (left) and in Tailless (right) strains. In FA Tailed strains the repeated fragments are separated by a 2.6 kb DNA sequence. (ii) DSB ends undergo 5' to 3' resection, creating 3' single-strand DNA tails. (iii) Annealing of single-strand DNA at complementary sequences creates an intermediate with mismatched base pairs from which (iv) the nonhomologous tails are removed by Rad1-Rad10 endonuclease with the assistance of Msh2–Msh3. Depending on how MMs are corrected, there are different possible outcomes of SSA: Mixed repair–some of the seven mismatches are corrected as F, some as A and some are not corrected (a); all MM are repaired as F (b); all MM are repaired as A (c); If mismatches are not corrected, progeny containing both genotypes will result in a sectored colony containing both alleles (d). B. Creation of one-tailed and pseudo-Tailless strains by Cas9 cleavage. C. Location of mismatches (MMs).

through the use of CRISPR/Cas9 (S1 Fig). We also gene-edited this construct to place the F and A segments immediately adjacent to each other (see Materials and Methods) such that the junction can be cleaved precisely by Cas9 (Fig 1A and S1 Table), without creating a nonhomologous tail (referred to as a Tailless strain). We also inserted the same "Tailless" Cas9 recognition site near the HO cleavage site to create a tailed version (Fig 2C).

We show that heteroduplex rejection strongly depends on the presence of the 3'-ended nonhomologous tails. The lack of nonhomologous tails led to more efficient SSA in both FA

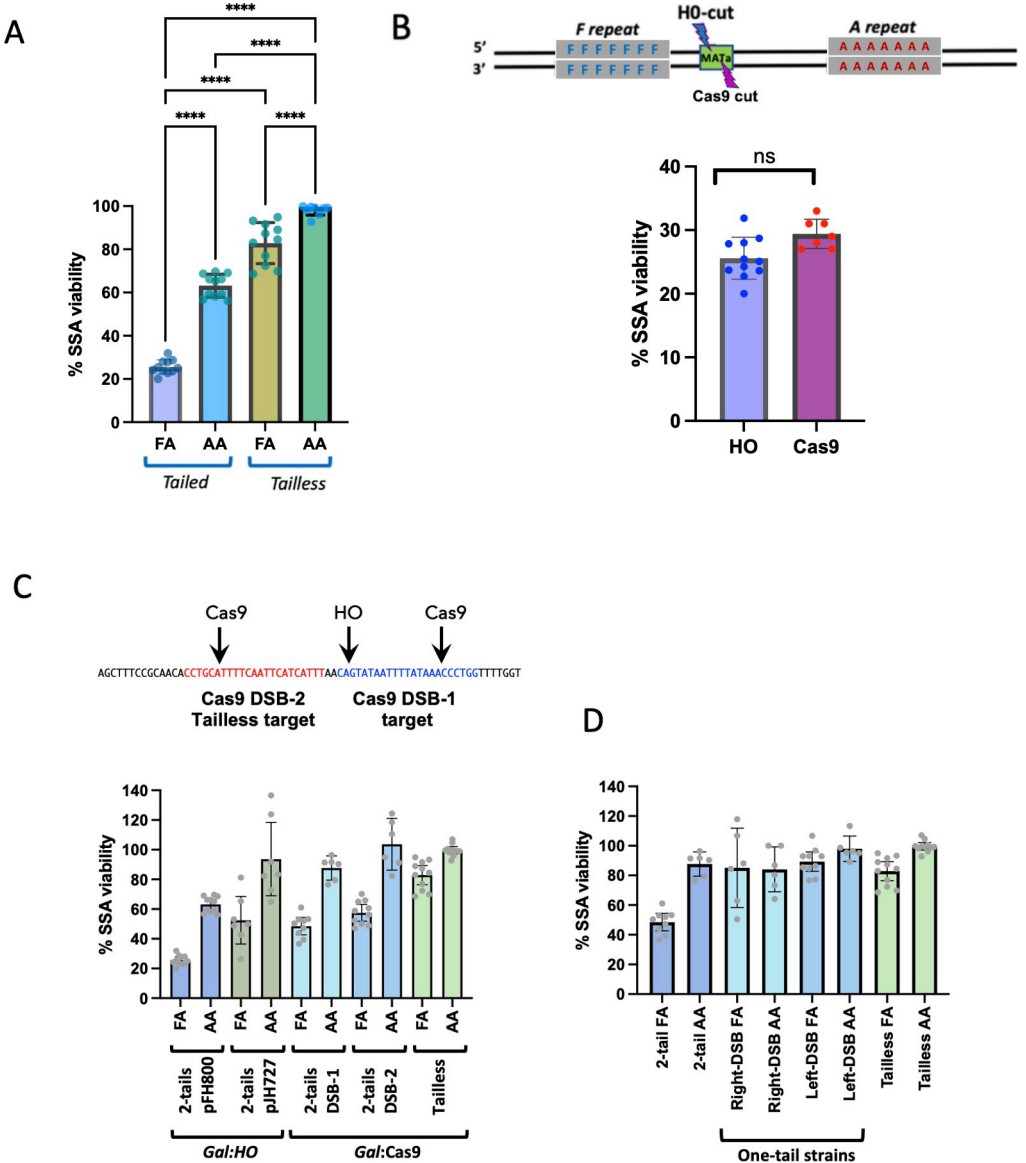

**Fig 2. SSA repair frequency. A**. Both mismatches and nonhomologous tails reduce SSA. The frequency of DSB-induced SSA was measured after induction of *GAL::HO* (in Tailed strains*)* or *GAL::CAS9* + gRNA (in Tailless strains) compared to cells grown on dextrose. Statistical test: One-Way ANOVA; Error bars represent the standard deviations; **** = p< 0.0001. **B**. Constitutively-expressed Cas9 was compared with *GAL::HO* induction of SSA in a tailed strain. Cas9 targets a site near the HO DSB (DSB-1 in panel C). Experiments were independently repeated at least five times. Error bars indicate standard deviations. Experimental means were statistically compared by the unpaired t-Test. **C**. Comparison of *GAL::HO* and *GAL::CAS9* induced SSA when Cas9 cleaves the same sequence used in the Tailless construct (DSB-2 in red). **D**. Strains with only one nonhomologous tail were compared with 2-tail and Tailless strains using *GAL::CAS9*.

and AA Tailless strains compared to Tailed strains. We show further that the pattern of mismatch repair changes dramatically in the absence of nonhomologous tails. When tails must be clipped off to complete SSA, mismatch correction favors retention of the allele on the opposite strand, producing a gradient of repair from the two ends. In contrast, in the absence of tails, assimilation of each mismatch occurred coordinately across the entire sequence. In this process a critical role is played by Msh2, which has multiple functions during SSA repair, including heteroduplex rejection, nonhomologous tail removal and mismatch correction.

In the course of this work we also discovered that the role of Rad52 in SSA is significantly affected by the presence of 3' nonhomologous tails. Whereas Rad52 is essential in SSA when there are such tails, a significant fraction of SSA events can occur without Rad52 in cases where there are no such tails.

## Materials and methods

### Strains and plasmids

Strain tNS1379 carries a duplication of 200-bp sequences upstream of the *URA3* open reading frame (designated AA) separated by 178 bp of pUC9 DNA, the HO endonuclease cut site (117 bp) derived from *MAT***a** [19], and 2.3-kb phage λ DNA. In tNS1357 (FA Tailed strain) the left-ward *URA3* repeat was replaced by sequences derived from strain FL100 and differs from the AA Tailed strain by six single-nucleotide substitutions and one indel within a run of 11 or 12 Ts [13] (Fig 1C and S1 Table). pFH800 [19] contains *GAL::HO*, *TRP1* and *CEN4* within a pUC19 plasmid. The *GAL::HO* plasmid, pJH727, was derived from YCp50 and carries *LEU2* and *CEN5*.

Tailless strains were obtained by gene editing tNS1379 (AA Tailed strain) and tNS1357 (FA Tailed strain). A DSB was created by Cas9 cleavage between the two repeated fragments and a 100-bp dsDNA designed to have 50 bp homology to the 3' and 5' ends of the left and right repeated fragments was used as a template for repair, allowing the removal of 2.6-kb phage λ DNA and other intervening sequences, resulting in a strain with two immediately adjacent repeats, designated as FA and AA Tailless strains yRT03 and yRT04 (S1 and S4 Tables).

Additional gene editing was employed to obtain an AF Tailed strain (YES49) (where the positions of the A and F segments are reversed), nFA (YES34) or AnF (YES65) Tailed strains (when the insertion of 1 T from the F segment was removed and the third base from mismatch 3 was changed to a G). DSBs were created by: pES6 (an *LEU2*-marked Cas9 plasmid that cuts upstream of the F repeat–used to create nF strains) in the presence of a ssDNA template (ES25) carrying the point mutation; pES11 (an HPHMX Cas9 plasmid that cuts downstream of the A fragment) was used to insert the F fragment obtained by PCR, into the tNS1379 strain, to obtain the AF Tailed strain (S4, S5 and S6 Tables)

To create strains in which there would be only one nonhomologous tail after Cas9 cleavage, PAM sequences were inserted near the 3' end of the left repeat (AGG) or the 5' end of the right repeat (CCG) such that a DSB was made at the exact junction between the repeats and spacer. pES8 (HPH-marked Cas9 plasmid that cuts upstream of the A repeat was used to insert a PAM adjacent to the right fragment), and pES18 (*LEU2*-marked Cas9 plasmid that cuts down-stream of the F repeat, was used to insert a PAM adjacent to the left repeat), in each case using a single-stranded oligonucleotide template to edit the site (ES67 and ES114-Right-PAM in S6 Table). The dsDNA templates used for editing were obtained by PCR amplification and the ssDNA templates were ordered from IDT (S6 and S7 Tables)

Deletions of *SGS1*, *MSH2*, *MSH6*, *MSH3*, *RAD51*, *RAD52*, *MLH1*, and *PMS1* were per-formed by replacement with KANMX, NATMX, or HPHMX cassettes [20] using PCR primers with 40-bp homology to sequences 5' and 3' to the open reading frame and 20-bp homology to MX cassettes [20]. Tailed and Tailless strains were transformed with the MX cassettes using high efficiency yeast transformation methods [21].

A novel inducible Cas9 plasmid was designed to generate two DSBs adjacent to Left and Right repeats in SSA models. The plasmid incorporates two gRNAs under the control of a Gal-inducible promoter. To generate pES53 an HPHMX plasmid that allows simultaneous expres-sion of two gRNAs, an additional gRNA scaffold expression cassette (amplified by PCR from pEZ165 plasmid [22]) was inserted into *Not1*-digested bRA66 (an inducible plasmid that

contains both *GAL*::Cas9 and a gRNA cassette) [23]. The Cas9 plasmid, denoted as pES55 and engineered for the generation of a targeted DSB proximal to the Right repeat, was constructed through the assembly of guide RNA primers ES201 and ES202 into the *Bpl*I-digested vector pES53, by T4 ligase. To generate pES56 plasmid designed for the induction of a DSB adjacent to the Left repeat, gRNA primers ES209 and ES210 were cloned into the *Not*I-digested pES53 plasmid using Gibson Assembly. The plasmid pES57, engineered for the concurrent induction of two DSBs, was synthesized through the incorporation of guide RNA primers ES201 and ES202 into the *Bpl*I-digested pES56 vector, employing T4 ligase for assembly.

pNSU319, for the transformation into the *rad52Δ* and the wild type strain, BY4741, was constructed by inserting the *CEN6-ARS* sequence from pRS316 into pNSU118 [24] between the *Bam*HI and *Sph*I sites by Gibson Assembly using oligos NS403-Cen and NS402-ura3 (S5 and S6 Tables).

## Analysis of SSA

HO endonuclease under the control of a *GAL10* promoter (*GAL*::HO), carried on a centromeric plasmid, pFH800 [25], was induced by plating cells onto YEP-galactose or YEPD plates to create a single DSB between the 200-bp repeated segments [13]. Colonies on YEPD and on YEP-Gal plates (DSB survivors) were replica plated on tryptophan drop-out plates (the selectable marker for pFH800) to determine SSA frequencies. After induction of *GAL*::HO nearly all survivors were SSA products and could thus be monitored by colony counting. Approximately 10% of cells lost the *TRP1*-containing plasmid. SSA repair was confirmed by PCR using primers NS154-pUC and URA3p14 upstream and downstream from the two repeated fragments (S6 Table) a 250 bp fragment indicates SSA repair; in the absence of DSB the PCR product is 2.9 kb. *GAL*::CAS9 strains were similarly induced using the appropriate marker, *LEU2* or HPHMX, carried on the *GAL*::Cas9 plasmid. The data for S3 Table were obtained by performing colony PCR on colonies obtained after galactose-induction of Cas9 cleavage, using the primers pUCp1, ura3p14 and NS408-102-1 (S6 Table). This combination of primers gives a 632-bp PCR product if the colony retains the parental configuration or 285 bp if an SSA product formed. In some cases both PCR products were present and were designated as "2 PCR products." Rare PCR products of other sizes, derived from *rad52Δ* strains, were not investigated.

For the *rad52Δ* transformation assay DNA from pNSU319 was prepared by digesting with 1) *Eco*RI 2) *Hin*dIII 3) *Sph*I and 4) *Hin*dIII + *Sph*I. Restriction enzymes were heat-inactivated. Equal quantities of digested DNA and uncut pNSU319 were transformed into BY4741 and the *rad52Δ* derivative from the Yeast Deletion Collection [26]. The number of transformants using the digested DNA was divided by the number of transformants from the uncut plasmid. The number of transformants were further normalized to the maximum transformation frequency from the samples digested with *Hin*dIII + *Sph*I transformed into BY4741.

## Constitutive Cas9 plasmids

pAB101, an HPHMX Cas9 plasmid targeting a site in the 117-bp HO cleavage site from *MAT***a** adjacent to the phage λ sequence [13] was used to create a DSB 268 bp from the left repeat. This Cas9 cleaves adjacent to the site cut by HO endonuclease (S1A Fig). gRNA targeting sequences are listed in S6 Table; pES02 is a HPH-marked Cas9 plasmid that cuts approximately in the middle between the two repeated fragments, while pES08 is an HPH-marked Cas9 plasmid that cuts 24 bp upstream of the right (A) repeat (S1A Fig). To create DSBs with only one nonhomologous tail we designed gRNAs to direct Cas9 cutting adjacent to the F fragment (pES19 –nonhomologous tail on bottom strand) or the A fragment (pES20—

nonhomologous tail on the top strand) Fig 1B. To investigate SSA frequency when DSBs were created at different locations within the spacer sequence, the FA Tailed strain was transformed with one of the plasmids, pAB101, pES2, pES8, pES19 or pES20, in the presence of a second, *LEU2*-marked plasmid to normalize for variations in transformation. bRA89 is a HPH-marked plasmid lacking Cas9 that was used as a control. Transformants were plated directly on *LEU2* drop-out plates (marker of the normalization plasmid) and replica-plated to HPH plates after three days of incubation at 30˚C. The survivors carrying both *LEU2* and HPH plasmids were counted. The number of colonies transformed with pAB101, pES2, pES8, pES19 or pES20 (DSB repaired colonies) were compared to the number of colonies transformed with bRA89 (no DSB). Plasmid concentrations were adjusted so that equal numbers of transformants were obtained using a strain lacking the relevant cut sites.

Experiments were independently repeated 5 to 9 times. Experimental means were statistically compared by a one-way ANOVA test.

## Analysis of mismatch repair after SSA

Genomic DNA was obtained from single colonies that were boiled in 50 μL of Lysis Buffer (10mM Tris-HCl, 2mM EDTA, 1% SDS). The 250-bp products confirming SSA repair were PCR-amplified using primers, NS154-pUC and URA3p14, (S6 Table) that were positioned 50-bp outside the repeated fragments. The amplifications were performed in 20 μL reactions (2XPCR FailSafe PreMix E from Biosearch Technologies, 1.25 U of GoTaq DNA Polymerase (Promega), < 1μg DNA template and 0.5 μM of each specific primers). The amplification consisted of an initial activation step at 95˚C for 5 min followed by 25 cycles, including a denaturation step at 95˚C for 0.30 min, annealing at 55˚C for 0.30 min, and extension at 72˚C for 1 min. The final extension was performed for 5 min at 72˚C. Individual PCR reactions were sequenced (Genewiz) using the reverse primer URA3p14, and the traces were aligned with the A and F sequence fragments. The chromatogram of the SSA repair products could show either the nucleotide from the A fragment or F fragment or the presence of both A and F alleles, suggesting that the MM wasn't repaired or that a colony derived from post-replicative cells in which repair occurred independently in two daughter nuclei. Approximately 50 samples were sequenced for each construct. Sequence analyses was carried out using the alignment feature of GENEIOUS 11.0.5 software.

## Strand linkage analysis–Barcode NGS

Sanger sequencing described above can show that one or more sites retained both F and A alleles, but cannot assign linkage of the markers. To determine which alleles were carried on a single DNA strand, we performed paired-end sequencing of individual SSA outcomes. Strand-specific analysis of SSA progeny was done as follows. We built six libraries: FA_WT Tailed, FA_*msh2Δ* Tailed, FA_*msh6Δ* Tailed, FA_WT Tailless, FA_*msh2Δ* Tailless, FA_*msh6Δ* Tailless. Each library contains sequences amplified from 96 SSA repaired colonies using 8 specific forward primers and 12 specific reverse primers (with 8-bp barcodes; S6 Table). Genomic DNA was obtained from single colonies as described above. Each independent SSA product was amplified for 25 cycles, using a unique pair of barcode primers in 20 μL of PCR mix using GoTaq (Promega) or Q5 polymerase (New England Biolabs). All 96 barcoded PCR products from each library were pooled, DNA purified, and analyzed using Amplicon-EZ, a Next Generation Sequencing (NGS) service, that uses Illumina Mi-seq technology (Azenta Life Sciences, Chelmsford, MA). The R1 and R2 paired-end reads were joined by BBMerge [27] at the highest stringency pre-setting. Length and quality score filtering (L>250 bp, Q>35, respectively) was done using FastP [28]. Mapping and quantification of sequence variants was done using

the Variable Antigen Sequence Tracer (VAST) [29] with the F-sequence as the reference and the A-sequence as the cassette. VAST variant_frequency reports contain frequency information for all observed recombinants between the reference and cassette(s) organized by colony and genotype. TSV files containing this information were processed using FA_variant_processor_MM2_fix.py, a Python script that labels MM positions as being either from the reference (F-sequence) or cassette (A-sequence) and treats sequences where the second position (MM2) does not agree with the 1st (MM1) and 3rd (MM3) positions as errors, and corrects them. For example, the FAFFFFF sequence is counted as if it were FFFFFFF. Each colony is then assigned a duplex genotype by picking the two most frequent sequences that have a frequency greater than 19%. Details and code are available from: https://urldefense.com/v3/__https://github.com/RandallTyers/FA_variant_processor__;!!DaRZpAeNFA!cln6dkQY1fClJSBLWZFoishnPpbKjirM-VBjkT86TJJE015Twh8XquhsibtUg0YEOaPqaOFAd2dLDw-iXLrabMI$ [github[.]com].

## Rationale for MM2 error correction

Mismatch 2 (MM2) is a single-base insertion/deletion in a run of 10 Ts. Poly(dT:dA) stretches as short as $T_{12}$ are notable for blocking DNA polymerases both in vitro and in vivo [30,31]. To establish the fidelity of identifying MM2 in repaired SSA events we analyzed DNA sequences derived from strains that either carried only F or only A alleles (i.e. MM2 was either in a run of $T_{12}$ or $T_{10}$). Using DNA polymerase Q5 High Fidelity DNA polymerase™ we found that 5 ± 2% of the reads appeared to be different from the template sequence in this region; such changes could come either from the PCR amplification or from subsequent steps in DNA sequencing. As has been previously reported [26] Taq polymerase generated single nucleotide indels at a much higher frequency (in our samples as high as 24 ± 2%). We note that the position of the indel within the polyT sequence cannot be determined. In analyzing SSA events between F and A alleles, in those instances when both MM1 and MM3 were derived from the same template, we assumed that MM2 was also retained/corrected in the same direction. The fraction of cases in which a given mismatch was changed to that of its neighbor is shown in S8 Fig.

## Data presentation

Figures were drawn in Powerpoint or in Prism. DNA sequence files for Fig 7 and viability data for other Figures are stored at https://zenodo.org/records/10498510.

## Results

### Nonhomologous tails stimulate heteroduplex rejection

We used a viability assay to evaluate the difference between Tailed and Tailless SSA strains after creating a DSB between the repeated fragments (Fig 1A). When the repeats were identical, viability was 68% in the AA Tailed strain and 100% in AA Tailless strains (Fig 2A), suggesting that the presence of nonhomologous tails derived from sequences between the repeated sequences inhibits repair. The 3% divergence between the repeats led to a decrease in repair frequency to only 26% in FA Tailed strains, 2.5-fold lower than the equivalent AA Tailed strain, whereas in the FA Tailless strain the viability was 85%. Hence there is some impact on viability imposed by the seven mismatches, but this is more pronounced in the presence of 3' nonhomologous tails that must be removed to complete SSA. Together, these results suggest that both the presence of nonhomologous tails and the mismatches (MMs) lead to heteroduplex rejection that causes a deficit in repair (Fig 2A).

Since the Tailed and Tailless strains were cut with different endonucleases, HO and Cas9, we constructed Cas9 Tailed strains to use as controls. The cut sites are shown in Fig 2C as

DSB-1 and DSB-2 where DSB-2 consists of the Cas9 target sequence from the Tailless strain inserted into the Tailed strain. Both Cas9 strains behaved similarly to the *GAL::HO* strain in that the FA strains underwent a greater degree of heteroduplex rejection when compared with the Tailless strains. We also observed that the strains with Cas9 had higher viabilities relative to the *GAL::HO* strain. The lower viability of the *GAL::HO* strain may be due to plasmid instability due to transcription through the centromere initiated by the bidirectional *GAL1-GAL10* promoter used by pFH800. Transcription through centromeres was previously shown to inactivate centromeres [32]. For comparison we also tested a *GAL::HO* plasmid, pJH727 where this does not occur and found that it had higher viabilities, comparable with the Cas9 plasmids (Fig 2C).

## Effect of one nonhomologous tail on repeat-mediated recombination

To investigate how only one nonhomologous tail impacts SSA repair and mismatch correction, we inserted PAM sequences adjacent to each repeated fragment in FA Tailed strains so that cleavage would occur exactly at one boundary of the repeats (Fig 1B). A Cas9-mediated DSB created adjacent to the left repeat results after annealing in a 2.6-kb nonhomologous tail (flap) from the bottom strand whereas a DSB created adjacent to the right fragment results in a 2.6-kb flap from the top strand. The viability of these one-tailed AA strains was comparable to the two-tailed strains and somewhat lower than the Tailless strain. In the FA strains, the presence of one nonhomologous tail yielded viabilities significantly greater than with the two-tailed construct, though still somewhat lower than the nearly 100% efficiency of the Tailless strain (Fig 2D). These results suggest that the presence of one non-tailed end is sufficient to allow more efficient strand annealing than with two tails and that heteroduplex rejection occurs significantly less often than in the two-tailed strain.

## Effect of mismatch repair genes on SSA in Tailed strains

To confirm and extend our understanding of the roles of the MMR machinery and Sgs1 helicase in SSA repair (a possible mechanism illustrated in S2 Fig), we deleted MMR genes and some helicases in the FA Tailed strain (Fig 3 and Table 1). As expected, *sgs1Δ* and *msh6Δ* increased SSA in the FA Tailed strain to nearly the level seen with AA, confirming that Sgs1 and Msh6 decrease SSA efficiency by promoting heteroduplex rejection [13,33]. Deleting *SRS2* resulted in very low viability in the FA Tailed strain (Table 1), consistent with previous studies that *srs2Δ* has a general defect in resolving recombination intermediates [5,34,35]. These results suggest that Sgs1 is the only 3' to 5' helicase required for heteroduplex rejection during SSA.

Although Msh2-Msh6 has been shown to be involved in heteroduplex rejection and *msh6Δ* increased the repair frequency of the FA Tailed stain two-fold, *msh2Δ* resulted in a significant decrease in viability, to 10% (Fig 3). Since Msh2 also forms a complex with Msh3 (MutS) and assists Rad1-Rad10 in removing nonhomologous tails [13], it is likely that the severe effect of *msh2Δ* compared to *msh6Δ* reflects its role in nonhomologous tail removal. Deleting *MSH3* also resulted in very low recombination (2.3%) in both FA and AA Tailed strains (Fig 3 and Table 1). Similarly, in *rad1Δ* Tailed strains the viability was only 1.5%. It is possible that the higher level of survival in *msh2Δ* (10%) versus *msh3Δ* or *rad1Δ* (2.3 or 1.5) could be explained by the fact that *msh2Δ*—but not *msh3Δ* could suppress heteroduplex rejection, thus increasing the number of survivors; however, while a *msh2Δ msh6Δ* double mutant yielded the same level of SSA as the *msh2Δ* mutant, a *msh3Δ msh6Δ* double mutant behaved as *msh3Δ* mutant, with only ~3% viability (S3 Fig). The deletion of MutL homologs (Mlh1 and Pms1) did not have a significant impact on viability in Tailed or in Tailless strains (Fig 3), confirming

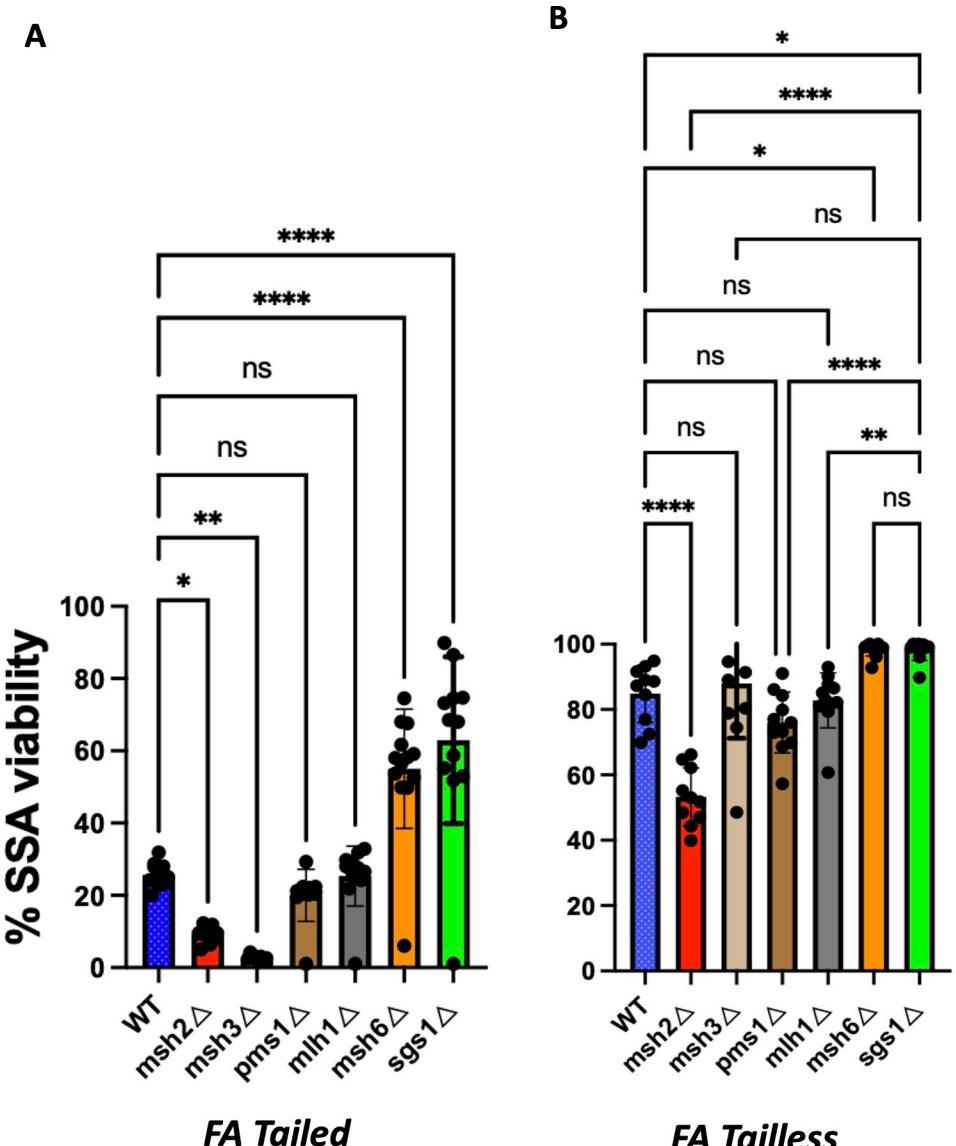

**Fig 3.** Effect of mismatch repair and helicase deletions on FA Tailed SSA induced with *GAL::HO* (**A**) and FA Tailless strains induced with *GAL*::CAS9 (**B**). Means of 5 or more independent experiments were statistically compared to WT Tailed or WT Tailless strains by One-Way ANOVA test. Error bars = SD. * p < 0.02; *** p = <0.001; **** p<0.0001.

our previous results that the Mlh1-Pms1 endonuclease is not required for heteroduplex rejection in SSA [13].

### Effect of mismatch repair genes on SSA in Tailless strains

The high efficiency of repair observed in Tailless strains (Fig 2A) can be explained by there being no need for Msh2/Msh3-assisted tail removal by Rad1-Rad10. However, in the FA Tailless strain the viability decreased modestly to 85% compared to AA Tailless, suggesting that the recognition of MMs by the Msh2-Msh6 complex might have an inhibitory effect on SSA repair. Indeed, deleting *MSH6* increased SSA repair efficiency of the FA Tailless strain to almost the level seen for the AA Tailless construct. However, *msh2Δ* decreased survival to 53%

**Table 1. Cell viability data collected from Gal inducible assay.** The effect of MMR proteins, helicases and recombinases on SSA repair efficiency. *GAL::HO* and *GAL::Cas9* were used in the Tailed and Tailless strains, respectively. The frequency of survival after an HO or Cas9 induced DSB was calculated by dividing the number of colonies growing on YEP–GAL by the number of colonies growing on YEPD. Each value represents the average from at least five independent experiments. One SD of the mean is shown in parentheses. ND = not determined.

| | % Viability | | | |
| --- | --- | --- | --- | --- |
| | Divergent repeats | | Identical repeats | |
| *Genotype* | **FA Tailed** | **FA Tailless** | **AA Tailed** | **AA Tailless** |
| *WT* | 26 *(3.1)* | 85 *(9.4)* | 68 *(5.0)* | 100 *(6.3)* |
| *sgs1△* | 68 *(12.0)* | 98 *(3.8)* | 100 *(3.2)* | 100 *(8.2)* |
| *msh6△* | 59 *(7.6)* | 92 *(3.2)* | ND | ND |
| *msh2△* | 10 *(1.9)* | 53 *(8.4)* | 3 *(1.6)* | ND |
| *msh3△* | 2.3 *(0.8)* | 88 *(16.7)* | 3 *(1.9)* | 97% *(10)* |
| *mlh1△* | 28 *(3.3)* | 81 *(8.4)* | 69 *(9.3)* | 91 *(8.4)* |
| *pms1△* | 22 *(2.8)* | 73 *(9.3)* | 57 *(11.0)* | 98 *(5.2)* |
| *exo1△* | 31 *(3.6)* | 50 *(6.1)* | ND | ND |
| *srs2△* | 0.8 *(0.0)* | ND | ND | ND |
| *rad1△* | 1.5 *(0.1)* | 61 *(5.8)* | 1.3 *(0.1)* | 77 *(6.8)* |
| *rad△* | 42 *(5.7)* | N.D | 72 *(7.6)* | N.D |
| *rad52△* | 0.2 *(0.0)* | N.D | 0.2 *(0.10)* | N.D |

(Fig 3 and Table 1), suggesting that there may be still another role for Msh2 in SSA involving divergent repeats. As expected, *sgs1Δ* also suppressed the effect of mismatches in the FA Tailless strain (Fig 3 and Table 1).

## The presence of nonhomologous 3' tails affects the repair of mismatches in SSA

In strains with 3% divergence between the repeated fragments, the resected 3' ssDNA tails will anneal to form a heteroduplex with six single-base mismatches and one "T" insertion distributed, as shown in Fig 1C and S1 Table. If these mismatches are not recognized and corrected by MMR proteins, they will exist as heteroduplex DNA and each strand will be used as a template for DNA replication so that both alleles will be found in SSA progeny as sectored colonies. If the MMs are corrected, then all descendants of the original cell will show the same genotype.

There is an ambiguity in the position of the insertion of a T (mismatch 2) within a run of 12 Ts. Because MM1 is also a T in in the F variant, there are actually 12 Ts in a row in F compared to 10 in the A variant. As we document below, there was a high level of discordance between the apparent repair of MM2 compared to the adjacent MM1 and MM3, likely attributable to errors in PCR amplification and sequencing. Hence, in most of the analysis below, we have only reported the fate of the 6 well-behaved mismatches, omitting MM2. By converting the indel to a base pair mismatch or by strand linkage analysis (below) we could also specify the fate of MM2.

We investigated by Sanger sequencing how MMs are corrected during SSA repair (Fig 4). For the FA Tailed strain, where SSA was initiated by inducing HO endonuclease, approximately half of the outcomes (66/143) displayed correction of all mismatched sites either as all F or all A (Fig 4A); of these, 57 (86%) repaired all sites as F. Among the remaining 77 colonies there was a mixture of fully corrected F or A alleles as well as a minority with unrepaired heteroduplex. Overall, the correction of the six mismatches displays a gradient of correction such that sites near the left end are corrected as F whereas sites closer to the right end showed a

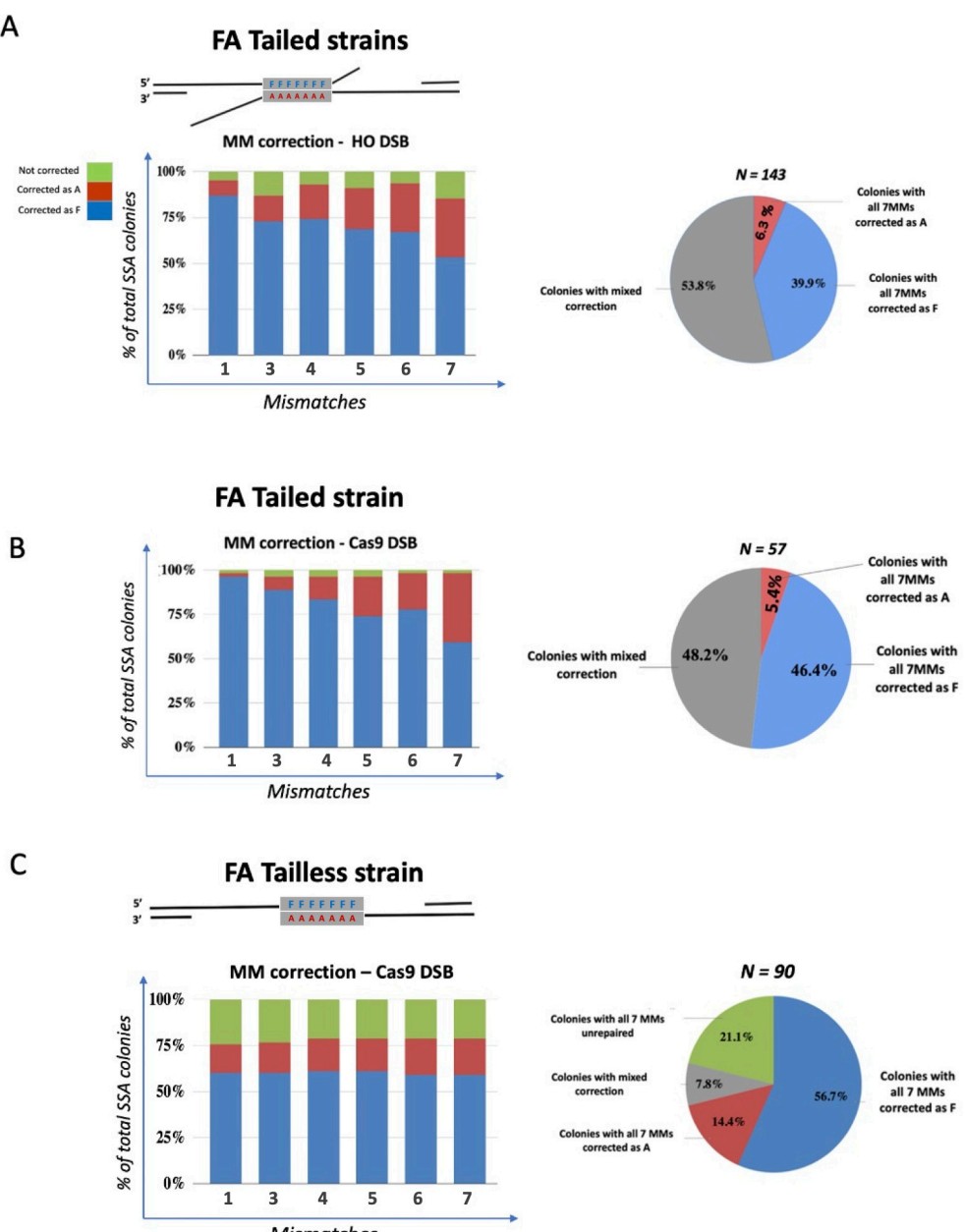

**Fig 4. Sanger sequencing analysis of mismatch correction in FA Tailed and Tailless strains.** Left: the six columns from the chart correspond to the repair of each MM except MM2 and represent the percentage of survivors that repaired a mismatch as A (red), as F (blue) or remained unrepaired (green). Right: overall distribution of SSA repair outcomes. **A**. Induction of SSA by expression of HO endonuclease. **B.** Induction of SSA after constitutive induction of Cas9 (pAB101) at DSB-1. **C**. Spectrum of repair products in the *GAL*::CAS9-induced FA Tailless strain.

higher proportion of corrections to A (Fig 4A). Among colonies with non-uniform correction MM1 was corrected to F in 87% (124/143) whereas MM7 was corrected to F only 53% (76/143) cases. We note that MM7 is about 50 bp further from the end of the homology than MM1, so that we likely underestimate the bias of correction at the right side of the repeat. Very similar results were obtained when we directed cleavage at almost the same site by introducing a constitutively expressed Cas9 plasmid pAB101 (Fig 2B and 2C). Again, among colonies

where there was non-uniform correction of markers, 96% (54/57) corrected MM1 to F, while only 65% (37/57) corrected MM7 to F. This difference is statistically significant (p<0.01).

We investigated the possibility that the indel located on the left end might cause the repair bias; however, we found comparable results in an nFA strain YES34 obtained from the FA strain by deleting the +T in the F allele and replacing the T at position 23 by a G, 5 bp upstream MM3 (S4A and S4B Fig and S1 Table). Again, there was a strong bias towards the repair of all alleles as F, with the highest level of F incorporation at the left end of the segment (S4C Fig).

The bias to correct sequences in favor of F is in fact a bias in favor of whichever allele is located close to the left end of the annealed structure. We created a Tailed SSA construct, designed as AF, in which the positions of the A and F segments are reversed (S5A Fig). When the A allele is located to the left of the DSB, repair now favors the A allele at the left end (S5B Fig), and, again, the replacement of the T deletion with a G to create a base-pair mismatch at MM2, 5 bp from MM3 (strain AnF), yielded a similar pattern (S5 Fig).

In contrast, in FA Tailless strains, Sanger sequencing analysis of 90 colonies revealed a remarkably uniform pattern of correction of all the mismatches (Fig 4C). Similar to the FA Tailed strains, where 40% corrected all markers to F and only 6% corrected all to A, in the Tailless strain all 6 scored MMs were corrected in ~ 56% of events as F while ~15% were corrected to all A. In FA Tailed strain, the majority (54%) of survivors are genetically heterogenous colonies, the results of a mixed MM correction. In this context, 'mixed MM correction' describes a scenario where mismatches are addressed through a combination of corrections: some are repaired using nucleotides from the top strand ('F'), some using nucleotides from the bottom strand ('A'), and there is a possibility that certain mismatches may remain uncorrected (Fig 1Aa). In FA tailless strain only 8% of the SSA survivors (7/90 colonies) showed a mixed correction (Fig 4B and 4C).

One notable difference from the results with Tailed strains is that in 21% of the FA Tailless colonies none of the 6 scored MMs was apparently repaired during SSA; these progeny presented a pattern where all of the sites contained both F and A alleles. Because we found the same spectrum of repair when the DSB was introduced by HO or by Cas9 cleaving at a nearby site (Fig 4A and 4B) the increased number of fully heteroduplex colonies in the Tailless strain is not an inherent consequence of inducing the Cas9 nuclease. However, the site being cleaved in the Tailless strain may be less accessible. It is possible that there is a delay in cleavage, such that a fraction of the colonies would have had independent DSB repair in the two daughter nuclei. In some of those cases one SSA event would yield only F alleles whereas in the second, only A alleles would be recovered, yielding what appeared to be persistent heteroduplex across the region. Alternatively, the mismatch repair machinery may be less efficiently recruited in the absence of nonhomologous tailed intermediates. In any case, it is clear that the spectrum of repair of mismatches in the Tailless strain is markedly different from that in the Tailed cases.

Evidence that *GAL::CAS9* cleavage at some locations might be less efficient, and possibly delayed, comes from a similar galactose-inducible Cas9 cleavage of the Tailed strain, cutting in the middle of the phage λ sequences, directed by plasmid pES2, i.e., at a site different from that shown in Fig 4 (S6 Fig). We found that 4/26 (15%) colonies analyzed showed the presence of both F and A alleles at each site. Among the other outcomes, we saw the same strong bias in favor of complete correction of alleles to F (23%) compared to full correction to all A alleles (4%) that we found with HO-induced cleavage and the same gradient of correction favoring F alleles at the left end and A alleles towards the right end. These data reinforce the conclusion that mismatch correction in the Tailless strain is quite different from the Tailed strain, that the presence of the 3' nonhomologous tail removal impacts the way mismatches are corrected in the SSA intermediate, and that this pattern can be seen even when Cas9-directed cleavage is less efficient than at other sites.

## Effect of DSB position on mismatch correction

To investigate how only one nonhomologous tail impacts mismatch correction, we used the one-tailed AF strains, analogous to the FA strains described above (S5 Fig). We note that in the AF arrangement, there is a bias toward recovering the A alleles (S5 Fig). Here, SSA events were induced by introducing a constitutively-expressed Cas9.

Sanger sequencing of individual colonies revealed that the spectrum of mismatch correction was different between the two one-flap cases (S7 Fig, compared to S5 Fig). With the flap on the left end of the annealed region there is the same strong preference for incorporation of the repeat sequence on the left side and a clear gradient when the DSB is made at the left junction, as we saw for the two-tailed strain (S5 Fig). However, the spectrum when a DSB is made at the right junction and the flap is on the top strand appears substantially different from that of the two-tailed strain and more similar to the uniform level of correction seen in the Tailless strain. The presence of an increased number of unrepaired heteroduplex DNA, especially for MM7, complicates the comparison. It is possible that the lack of a tail on the left end of the annealed region eliminates alternative ways of correcting mismatches near the left end of the sequence and thus leads to a more uniform pattern of correction at all sites.

One possible way that markers near the left end would be corrected in favor of the top strand would be if the step of filling-in the gap on the bottom strand by DNA polymerase δ (Fig 1A) frequently involved the 5' to 3' proofreading activity of the polymerase, resulting in a removal of some of the heterologies on the bottom strand. We previously showed that the proofreading 3' to 5' activity of DNA polymerase δ can resect as far as 40 nt from the 3' invading end after Rad51-mediated strand invasion [13,18,36]. A similar 3' to 5' removal of the 3' end after flap removal would also favor incorporation of the sequence on the top strand. When a DSB is created adjacent to the right fragment, leading to one nonhomologous tail on the top strand, the most proximal mismatch, MM7 (localized 70bp downstream the 3' tail), would be less likely to be corrected (S7B Fig).

To assess whether Polδ proofreading activity accounted for the strong directional correction of markers close to the left end, we introduced the *pol3-01* proofreading-defective mutation that we had shown resects the 3' end after strand invasion [18]. As shown in Fig 5A and 5B the pattern of mismatch correction in the two-tailed strain was unaltered in the *pol3-01* mutant. Thus, 3' end-removal by proofreading is unlikely to be the source of the strong bias in repairing markers near the left end; but it is possible that flap removal could sometimes include part of the homologous sequences (Figs 5 and 6).

## Effect of mismatch machinery and helicase mutations on SSA repair

As previously described [13] *msh6Δ*, but not *pms1Δ* and *mlh1Δ*, increased the level of SSA in the Tailed FA strain compared to WT strain (Table 1 and Fig 3). Sequencing results, showed, as expected, all three mutations led to a strongly increased percentage of uncorrected MMs in the Tailed FA strain (Fig 7). This high percentage of heteroduplex was observed from MM3 to MM7 (Fig 7), suggesting that repair of these MMs is mostly dependent on the MMR machinery. However in *msh6Δ*, and *mlh1Δ* MM1 still showed a significant percentage of mismatch correction in Tailed strains. As this MM is located close to the extremity of repeated fragment, position +17 bp, (S1 Table), it is possible that its correction, and perhaps also some of the corrections of MM2/3, can occur by an alternative mechanism (see below).

As we noted above, Msh3 and Msh2 both play important roles in the clipping of 3' nonhomologous tails; deletion of either gene reduced the efficiency of SSA to ≤10%. Among the low-frequency SSA events that occur in the absence of Msh2 the percentage of heteroduplex (unrepaired MMs) was very low (<20%) (Fig 7A). Moreover, the pattern of correction for *msh2Δ*

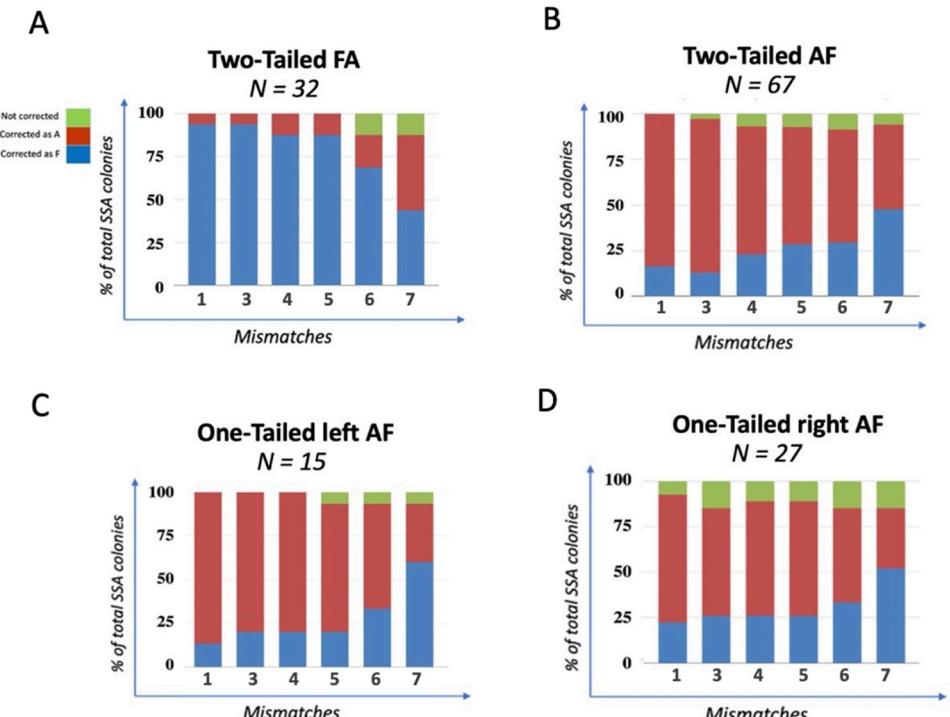

**Fig 5. Distribution of MMs corrected as F or A in *pol3-01* mutants.** Sanger sequencing analysis of SSA progeny showing the polymerase δ effect on mismatch correction during SSA repair in: FA and AF Tailed strains. DSBs were created between the two repeated fragments (**A & B**), adjacent to Left Repeated Fragment (**C**) or to Right Repeated Fragment (**D**).

was remarkably different than wild type for MM4, MM5, MM6 and MM7, as they were now repaired strongly in favor of the alleles on the bottom strand (A fragment). These data suggest that during SSA in the absence of Msh2 there is a small population of cells that might undergo repair through an alternative mechanism of nonhomologous tail removal. These SSA outcomes are characterized by a transition that leaves MMs 1–3 mostly unrepaired while MMs 4–7 are repaired as A.

We then examined the roles of the mismatch repair genes and helicases when SSA occurred in the Tailless strain (Fig 7B). In the absence of nonhomologous tails, deleting *MSH6* an*d PMS1* led to the great majority of repair events retaining heteroduplex DNA at each marker, while *sgs1Δ* had little effect. Strikingly, however, in the FA Tailless strain, deleting *MSH2* yielded 90% fully sectored colonies (i.e. where no site was repaired), a result quite different from the outcomes in the Tailed strain where there was strongly directional repair of the all markers. In the Tailless strain, there is no requirement for 3' tail clipping; consequently *msh2Δ* behaved quite similarly to *msh6Δ* or *pms1Δ* (Fig 7B). These data again suggest that the presence of nonhomologous tails affects mismatch correction. These data also demonstrate that there is a tail-dependent and Msh2-independent pathway, though inefficient, that leads to the recovery of SSA products with mismatch correction.

## Further characterization of mismatch correction by strand linkage analysis

To confirm the pattern of correction of the seven mismatches from heteroduplex DNA, we turned to Next Generation DNA sequencing of 96 independent SSA events as described in the Materials and Methods. The analysis of all these NGS reads (see Materials and Methods)

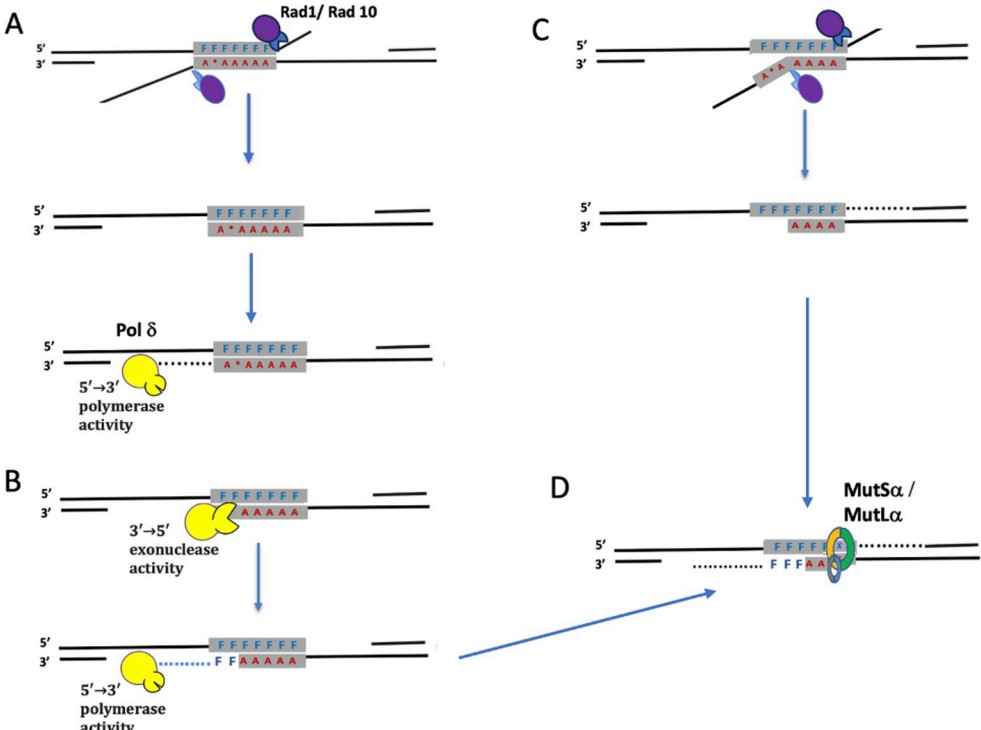

**Fig 6. Pathways leading to mismatch correction of heteroduplex DNA formed during SSA. A.** Heteroduplex DNA between the F and A variants has six single base-pair differences (F/A) and an insertion (*) that lies in a run of 12 Ts. Strand annealing is followed by Rad1-Rad10 clipping of 3'-ended nonhomologous tails. The 3' ends can then be used as primers for Polδ DNA polymerase to fill in the gaps. **B**. In some cases Polδ's 3' to 5' proofreading exonuclease activity may be stimulated to chew back the 3' end and remove heterologies as far as 40 nt from the end. Filling in by the same polymerase would lead to apparent mismatch correction of the terminal heterologies. **C**. Mismatches close to the 3' end might also be excised by Rad1-Rad10 if the 3' end is not firmly paired to its complementary strand. **D**. Mismatch repair performed by MutSα and MutLα is likely biased to remove mismatches on the strand containing the filled-in (and perhaps not yet ligated) strand, creating an opposite gradient of repair near each end.

allowed us to characterize more completely the repair events occurring in each colony (see Fig 8). In particular, we could unequivocally score the fate of MM2. Strand-specific analysis showed which alleles were found in a given repair event and revealed which alleles were in tandem when some sites were uncorrected (Figs 8, S8 and S9). In most cases, MM2 was corrected as the adjacent MM3 and when MM3 was uncorrected (i.e. both F and A alleles were present in a single colony, MM2 was also uncorrected; but there is still a high level of discordance for MM2 that may reflect sequencing problems (S9A and S9B Fig).

The most representative genotypes of six different libraries of SSA progeny, obtained from Tailed and Tailless FA WT, FA *msh2Δ,* and FA *msh6Δ* are shown in Fig 8. In WT strains the most abundant pattern of correction showed a strong affinity for the F (left) repeat. Overall, this analysis parallels the results from Sanger sequencing (Fig 4). For example, in the Tailless WT strain, about 1/3 of the outcomes were unrepaired heteroduplexes, 40% were fully repaired to F and about 22% fully repaired as A, with very little evidence of biased repair of the markers closer to the left end. In the WT Tailed case, there was an increased proportion of correction of markers at the left end, as we saw in the Sanger sequences, but there were more instances of unrepaired markers than we had seen in Sanger sequencing.

Consistent with the results presented above, the highest percentage of sectored colonies (arising from post-repair replication of heteroduplex DNA) was observed in Tailless strains

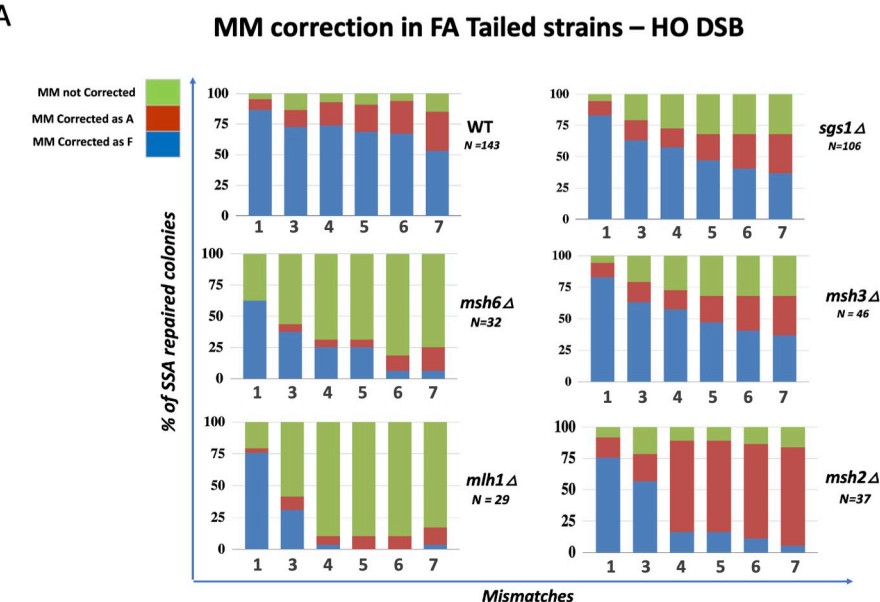

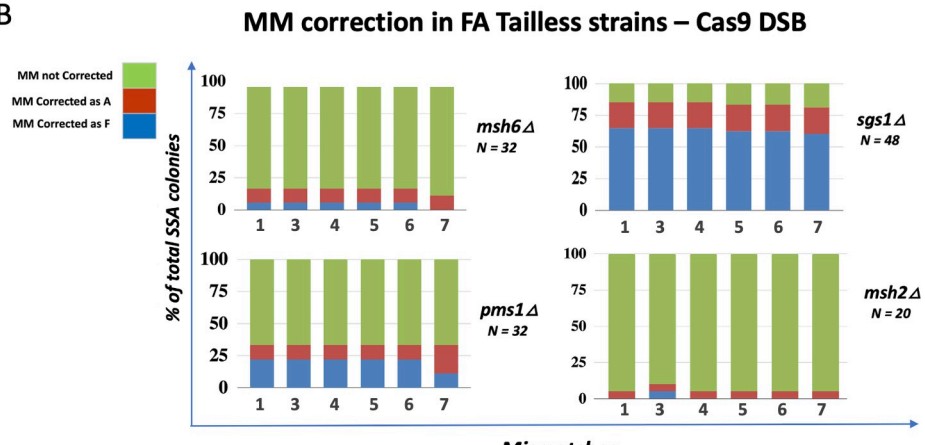

**Fig 7.** A. Impact of MMR machinery on MM correction during SSA in FA Tailed and Tailless strains. **A**. Mismatch correction in Tailed strains. Heteroduplexes (unrepaired MM) are shown in green. **B**. Effect of MMR mutants in FA Tailless strains.

*msh2Δ* and *msh6Δ* (∼ 60% and ∼ 40%, respectively), where none of the 7 MMs was corrected/repaired. Some of these heteroduplex outcomes likely represent failures of mismatch repair rather than any delay in cleavage. Unlike the Tailless strains, *msh2Δ* and *msh6Δ* Tailed strains resulted in an increased number of colonies with a mixed pattern of correction—where only some of the seven MMs were corrected. For example, in *msh6Δ* Tailed strain, the most representative genotype showed heteroduplex for mismatches MM3 through MM7 while the first two mismatches were repaired as the F alleles, again consistent with the results obtained by Sanger sequencing and suggesting that markers near the left end of the fragment may be repaired in a MMR-independent fashion. As we found with Sanger sequencing, the *MSH2* deletion in the FA Tailed strain produced a pattern of correction quite different from either WT or *msh6Δ* strains (Fig 8).

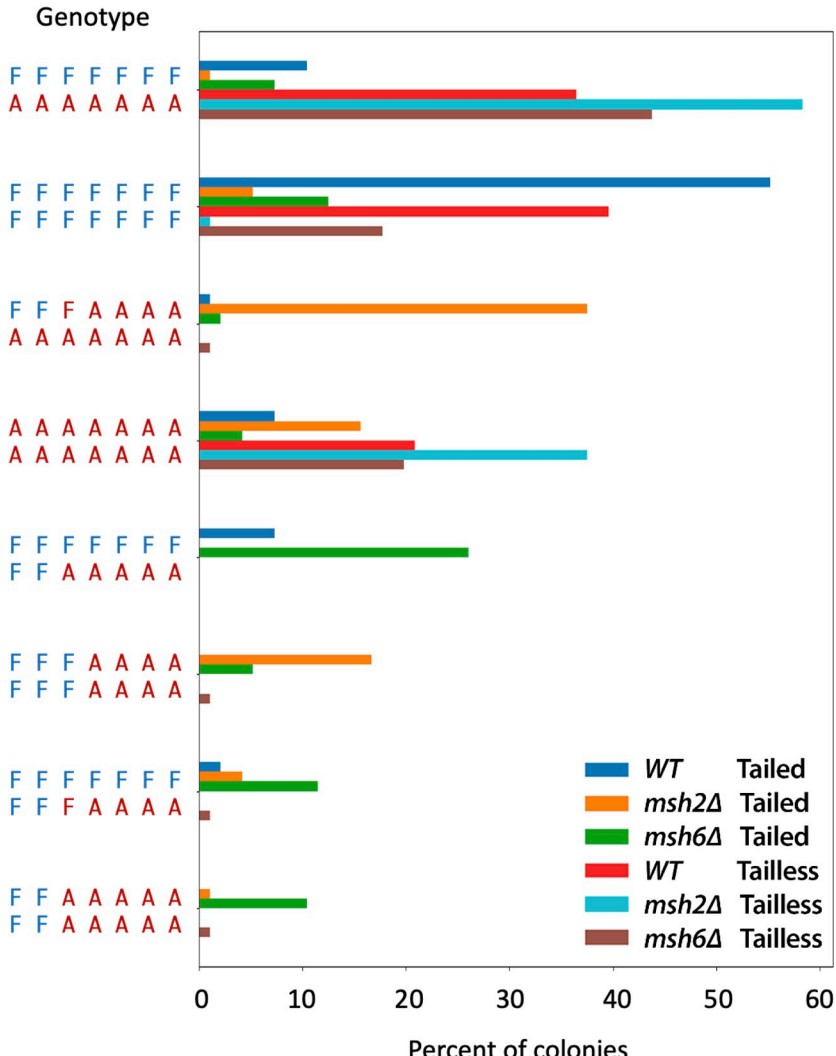

**Fig 8. Representative genotypes of SSA repaired progeny analyzed by strand linkage analysis.** The eight most frequent patterns of MM correction were identified by analyzing all reads for each of 96 colonies per library and assigning them a genotype based on the most frequently observed read(s); see Material and Methods for details. Statistics for the number of reads per colony: mean ± sd = 186 ± 112; Interquartile range (IQR) = 111–241; Range = 7–1139. Only genotypes that appear more than 9 times for at least one of the strains are shown; complete results are shown in S8 and S9 Figs.

This strand-specific analysis also shows that in an *msh2Δ* Tailed background there is often partial MMR with the primary product being heteroduplex at the first 3 sites but fully corrected at the remaining 4 sites: FFFAAAA/AAAAAAA. Furthermore, the next two most abundant products are colonies with only FFFAAAA or only AAAAAAA. It is possible that a cell with the partially corrected heteroduplex arrangement (FFFAAAA/AAAAAAA) then replicated, but only one of the two daughters survived; this might occur if one of the two strands of the SSA product did not complete tail processing (as Msh2 plays an important, but not essential, role in that process) and only one strand was ligated prior to replication. Alternatively, there could be more than one round of mismatch correction before cells replicated.

While the *msh6Δ* Tailed strain produces many different products, the primary one (FFFFFFF/FFAAAAA) is the inverse of the *msh2Δ*, with FFFFFFF/FFFAAAA also being

abundant. Moreover, similar to Tailed *msh2Δ*, the next most abundant outcomes appear to be the result of either another round of mismatch repair before replication or failure to recover both replication products, yielding only FFFFFFF, FFAAAAA or FFFAAAA.

In addition, this analysis reveals that even the WT Tailed strain produces a heterogeneous pool of products with about 10% showing a partial repair of one strand and more than 10% having no evidence of repair.

## *RAD52* becomes dispensable for SSA without 3' nonhomologous tails

Initially, we deleted *RAD52* to confirm that Cas9 was cleaving efficiently, on the assumption that SSA requires Rad52. Indeed in tailed strains, when a DSB was made by *GAL:HO*, the survival frequencies in a *rad52Δ* background using the FA and AA substrates were 0.04% or 0.14%, respectively. *GAL::CAS9 rad52Δ* strains similarly showed a decrease in viability relative to wild type strains (0.43 and 1.1% survival for Cas9 DSB-1 and 6.6 and 8.3% for Cas9 DSB-2, for the FA and AA strains respectively). PCR analysis revealed the presence of survivors with SSA products as well as those that retained the parental configuration. Colony PCRs from the latter group (*GAL:: HO* and Cas9 DSB-1) were sequenced (47 colonies in total). These were found to have mutated cut sites as a result of imprecise end joining which would prevent HO or Cas9 from re-cutting those sites. Although SSA products were present in *rad52Δ* strains, the overall frequencies of SSA were very low compared to those seen in Tailless strains. When Tailless strains were examined significant fractions of *rad52Δ* colonies were viable in both the FA and AA strains (43% and 35% respectively) (Fig 9). Upon examination of individual colonies by PCR, most (198 of 203 colonies) had SSA products; although, a small fraction, about 4% in the FA strain, contained the original SSA substrate. Four of these PCR products were sequenced and were determined to be products of imprecise end joining. Hence the higher viability cannot be attributed to an inability of Cas9 to create DSBs in these strains or to a highly elevated level of nonhomologous end-joining.

Rad52 was also largely dispensable in strains where SSA involved only one long nonhomologous tail; as shown in Fig 9, these one-tail *rad52Δ* strains showed viabilities between 8 and 43% and between 62% and 79% of the colonies contained SSA products, as determined by colony PCR analysis (S3 Table). Most of the remaining colonies possessed either only the fulllength (i.e. unrearranged) substrate or a mix of the two. Six colonies where the DSB was made at the left junction and which possessed the full-length substrate were examined by sequencing. All contained mutations of the target sequence as a result of imprecise end joining. Finally, we created a pseudo-Tailless strain by creating two Cas9 cuts, leaving the two repeats without nonhomologous tails. Here, too, there was significant SSA in the absence of Rad52 (Fig 9). That Rad52 was not required for SSA between 200-bp repeats in the absence of nonhomologous tails was surprising. Previously, Rad52-independent SSA had only been surmised in deletions of very long, repeated genes such as within the 9-kb ribosomal DNA array, often deleting more than one repeat [37].

To generalize this observation we employed a DNA transformation assay that was independent of Cas9 DSBs. In this experiment a *URA3*-marked centromeric plasmid was constructed using the same 200-bp repeats and the phage λ spacer sequence used in our SSA strains (Fig 10). Plasmid DNA was cut with restriction enzymes, *Hind*III and *Sph*I, so that the spacer sequence was removed and the repeats were adjacent to the ends of the DNA, in direct orientation, which would allow the creation of a *URA3* circular SSA product. Colonies were analyzed by PCR to confirm that SSA had occurred (S3 Table). In addition to this "no-tail" sample, DNA samples were prepared where one-tail was present (digested with either *Hind*III or *Sph*I) or two tails were present (*Eco*RI digested). When transformed into a *rad52Δ* strain, DNA with no-tails transformed at a significantly higher frequency than DNA with two-tails.

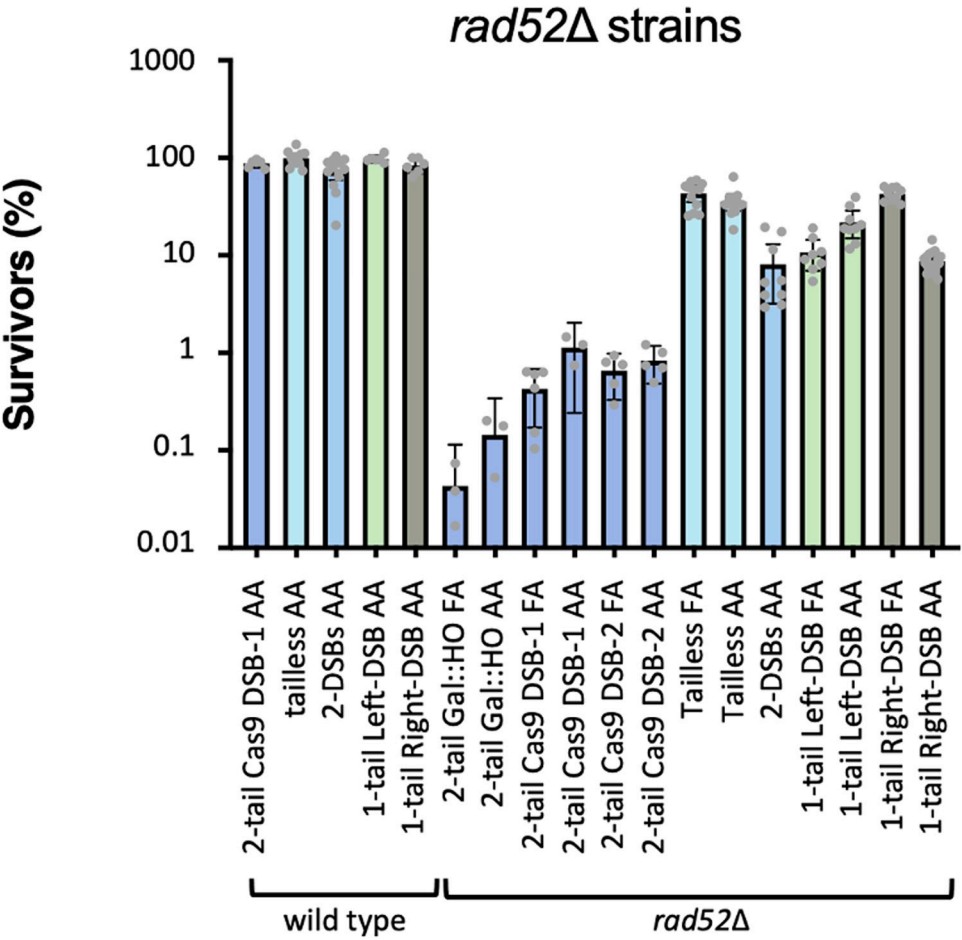

**Fig 9. Effect of *rad52Δ* on SSA.** Survival frequencies *rad52Δ* and wild type strains using *GAL*::CAS9. The graph shows the results of 1-tail, 2-tail, tailless as well as a strain where Cas9 cuts both junctions between the repeats and the spacer DNA (2-DSB strain diagrammed in Fig 1B).

## Discussion

In our study we constructed an SSA substrate that does not produce nonhomologous tails after cleavage by Cas9. With this substrate we could directly show that nonhomologous tails are important for heteroduplex rejection as both AA and FA Tailless strains have similar viabilities after DSB-induction. Although largely free from heteroduplex rejection, there is a small, but statistically significant drop, 17%, in viability in the FA strain, indicating that some heteroduplex rejection can occur in the absence of tails. Furthermore, this heteroduplex rejection can be suppressed by *msh6Δ* and *sgs1Δ*, whose role is also seen in the tailed strains [13,38].

Previous work by Chakraborty et al. [33] suggested that nonhomologous tails are required for heteroduplex rejection based on the observation that overexpression of Msh6 led to an increase in heteroduplex rejection, possibly by reducing the abundance of Msh2-Msh3 complexes needed for tail removal. In an opposing sense, Msh3 overexpression led to a decrease in heteroduplex rejection. The authors proposed that nonhomologous tails, as well as Msh6, are important for recruiting Sgs1 to enable heteroduplex rejection and once the tails have been clipped, MMR can proceed.

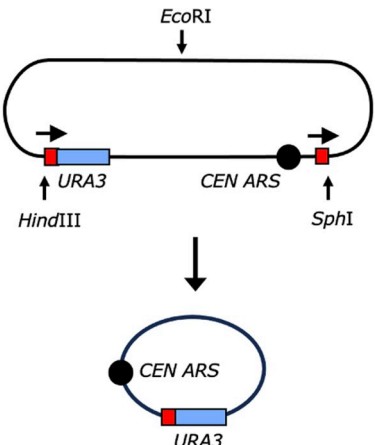

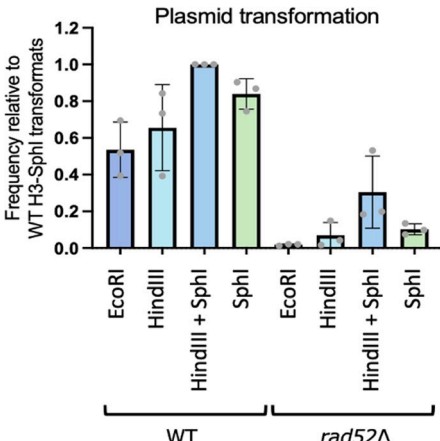

**Fig 10. Plasmid transformation assay in *rad52Δ* and wild type strains.** A plasmid containing 200 bp repeats, *URA3*, and *CEN6* was transformed into wild type and *rad52Δ* strains. The plasmid was cut with the restriction enzymes shown to mimic one-tail, two-tail or tailless SSA. Survival frequencies are shown for the transformation normalized to the uncut plasmid and to the highest transformation frequency, obtained when the DNA was cut with *Hind*III + *Sph*I.

Mlh1 and Pms1 had little effect on viability compared to wild type in the Tailless strains which is similar to results in the Tailed strains. Previously, *mlh1* mutants in budding yeast were found to increase the frequency of non-crossover events in an ectopic gene conversion assay [16], unlike SSA where *mlh1Δ* does not affect SSA frequencies. This implies that heteroduplex rejection of the annealed intermediate in Synthesis Dependent Strand Annealing differs from that in SSA. Hum et al. [16] proposed that the presence of nonhomologous tails during SSA allowed annealed intermediates to escape heteroduplex destruction initiated by Mlh1. Our observations that yeast Mlh1 does not play a role in heteroduplex rejection in the Tailless or tailed strains indicates that a factor other than the presence of tails plays a critical role in determining whether heteroduplexes are retained.

Curiously *msh2Δ* showed a decrease in viability in the Tailless strains that would lack the junctions that Msh2-Msh3 complexes recognize and bind. Furthermore, *msh3Δ* did not have this effect. This indicates that Msh2 may have another function outside of its role in recognizing the junctions in the annealed intermediate, although the pattern of mismatch repair of *msh3Δ msh6Δ* was comparable to that seen in *msh2Δ* and very different from that seen with either *msh3Δ* or *msh6Δ* alone.

Another interesting observation is that even in AA Tailed strains, where the repeated fragments are identical, there was a significant increase of viability from 68% to 100% when *SGS1* was deleted (Table 1). This finding suggests that recruitment of Sgs1 and its associated proteins to heteroduplex DNA can be independent of the presence of mismatches themselves and may reflect recruitment of Sgs1 by the Msh2-Msh3 complex that recognizes the branched structure created by the presence of the 3' nonhomologous tail [13,15,33] (S2 Fig). It is possible that even in the absence of mismatches, the inherent possibilities of misaligning the region of 10 Ts could provoke the attention of Msh2-Msh6. Previously it was reported that Msh2-Msh6 will coimmunoprecipitate with Sgs1 which is consistent with Sgs1 being recruited with either Msh2 or Msh6 [15,33,39].

Given that the two-tailed strains experienced heteroduplex rejection that the Tailless strains largely escaped we were also interested in the properties of one-tailed strains. We found that these one-tailed strains were nearly as efficient in SSA as the Tailless strains, for both the AA and FA substrates. These results suggest that one annealed end, lacking a tail, is sufficient to overcome most heteroduplex rejection. It is possible that the end lacking a tail can be used to initiate fill-in DNA synthesis quickly enough to stabilize the intermediate and allow strands that were unwound by the rejection machinery to be re-annealed.

## 3' flap structure influences the bias of mismatch repair

The data presented here, along with previous studies [13] show that Msh2-Msh6, but not Pms1-Mlh1, are required for heteroduplex rejection; however, the correction of the mismatches themselves requires both Pms1 and Mlh1. Our data suggest that the decision to correct a given mismatch in favor of one allele or the other is influenced by the presence of a 3' nonhomologous flap, which favors the retention of the sequence on the 5' end of the strand opposite of the flap. As discussed earlier, it is unclear when mismatch correction occurs relative to the Msh2-Msh3 aided Rad1-Rad10 mediated flap removal, but heteroduplex rejection most likely occurs before 3' tail clipping [39]. We suggest that flap cleavage precedes mismatch repair and that the newly-created 3' end after flap removal may serve as the guide to repair mismatches favoring the opposite strand (Fig 6). Possibly the cleaved end would be perceived in the way that a nicked strand is designated to be the corrected strand during DNA replication [40]. The lack of effect of *pol3-01* apparently rules out removal of the 3' end of the annealed structure as the basis of the strong bias in correction of the left end. We note that there is no such increase in the correction of MM1 in the tailless strain, implying that the processing of the tail itself plays a role in triggering this biased correction.

Recently the influence of the length of the 3' nonhomologous flaps in directing mismatch repair during SSA in mammalian cells has been addressed. In a model with a single mismatch in a 152-bp segment, SSA events exhibited a modest 60–80% bias in correcting the mismatch prompted by the shorter of the two nonhomologous tails (33 nt vs. 11 nt) and that the pattern is reversed when cleavage creates tails of 0 and 42 nt [41]. Another recent study by Trost et al. [42] found that mismatches proximal to a nonhomologous tail were preferentially removed. However, unlike in budding yeast, MLH1 as well as MSH6 caused a reduction in SSA outcomes with mismatched substrates. In yeast, Mlh1 and Pms1 are needed for mismatch correction *per se* but showed no effect on heteroduplex rejection in SSA. Trost et al. also examined the bias in repair when one nonhomologous tail was 268 nt long and the other varied from 16 nt to 9100 nt [42]. Longer tails impaired SSA both for perfectly matched 267-bp repeats or for 1–3% divergent sequences. A marked gradient in repair, favoring retention of the allele on the strand opposite the nonhomologous tail, proved to be independent of varying the length of the second nonhomologous tail.

There seems to be some intrinsic bias in the region that strongly selects the left repeat (whether F or A) to be used as the template for correction. This bias is especially evident for MM1-MM3. It is possible that the presence of both an indel at MM2 and the mismatch at MM3 could create a mismatched site that is recognized more strongly than other sites and influences the repair outcome; however, when we replaced the indel with a T:G mismatch, separated by 5 bp from MM3, the overall spectrum of mismatch correction was unchanged (S4 and S5 Figs).

The pattern of mismatch repair suggests that in Tailed strains there are often partial repair events that terminate (or start) near MM3 resulting in "patchy" mismatch incorporation. However, in Tailless strains, the assimilation of each mismatch occurred at the same frequency at each location. We note also that there is still a bias in favor of retaining the sequences of the left repeat, whether it is F or A. This result suggests that sequences adjacent to the repeats exert some effect on the repair process; but at this time we do not know what DNA sequence features, alterations in chromatin structure or effects of nearby transcription might be. The repair often seems to be "all or none" across at least the sequence. These results imply that there is a broad excision/repair event that covers most of the region. Whether mismatch repair happens before the DNA is filled in and ligated is not known.

In both Tailed and Tailless strains, Msh2 was found to play a central role, as this protein, depending on its partners, is involved in critical steps during SSA repair: heteroduplex rejection, nonhomologous tail removal and mismatch correction. We note that in contrast to wild-type and *msh6Δ* mutants, the viability of *msh2Δ* Tailed strains is very low; moreover, the pattern of mismatch correction is almost the opposite of what occurs in *msh6Δ*. We suspect that there is another, low-efficiency pathway that can operate without Msh2 that would account for these unusual outcomes.

## Rad52-independent SSA

The observation that SSA is partially independent of *RAD52* in Tailless strains was surprising for such short flanking repeats. Previously Ozenberger and Roeder [37] found Rad52-independent deletions involving much longer rDNA. Presumably the greater amount of homology or the repetitive nature of the rDNA array allows recombination to proceed without Rad52. In this study we examined the opposite extreme where the amount of homology was limited to only 200-bp. Rad52-independent SSA may be possible because as resection begins there is a greatly reduced number of sequences that need to be searched before the complementary sequences are found. Alternatively or additionally, the close physical proximity may promote *RAD52*-independent annealing. When we in effect moved the repeats apart by creating two DSBs so that the tailed strain became tailless, we observed a decrease in viability relative to the Tailless strain suggesting that close physical proximity is important. SSA has also been shown to involve another strand-annealing protein, the Rad52 paralog, Rad59, especially when the repeats were only a few hundred base pairs [17,38]. It is likely that Rad52-dependent SSA still requires Rad59.

## Supporting information

**S1 Table. Genomic sequences of the repeated fragments.**
(DOCX)

**S2 Table. Comparison of inducible and constitutive DSBs.**
(DOCX)

**S3 Table. PCR assay of colonies to assess completion of SSA.**
(DOCX)

**S4 Table. Yeast strains.**
(DOCX)

**S5 Table. Plasmids.**
(DOCX)

**S6 Table. Oligos and primers.**
(DOCX)

**S7 Table. Primers to assembly gRNAs.**
(DOCX)

**S1 Fig. Strand asymmetry impact on SSA repair.**
(TIF)

**S2 Fig. A model showing heteroduplex rejection and nonhomologous tail removal in an FA Tailed strain.**
(TIF)

**S3 Fig. Heteroduplex rejection and nonhomologous tail removal in the FA Tailed strain.**
(TIF)

**S4 Fig. Effect of replacing the indel at MM2 with a mismatch in the FA Tailed strain.**
(TIF)

**S5 Fig. Mismatch correction in AF Tailed strains.**
(TIF)

**S6 Fig. An apparent delay in Cas9 cutting impacts SSA outcomes.**
(TIF)

**S7 Fig. Nonhomologous Tail Removal Impacts MM Correction.**
(TIF)

**S8 Fig. Sequence analysis of 96 SSA events in different conditions.**
(TIF)

**S9 Fig. Concordance of mismatch correction between adjacent mismatches in different genetic backgrounds.**
(TIF)

## Acknowledgments

We are grateful for comments on the manuscript from Jeremy Stark and members of the Haber lab. Rebecca Tsai helped in the construction of the Tailless strain.

## Author Contributions

**Conceptualization:** Elena Sapède, Neal Sugawara, James E. Haber.

**Data curation:** Elena Sapède.

**Formal analysis:** Elena Sapède, Randall G. Tyers, Yuko Nakajima, James E. Haber.

**Investigation:** Elena Sapède, Mosammat Faria Afreen, Jesselin Romero Escobar.

**Methodology:** Elena Sapède, Neal Sugawara, Randall G. Tyers.

**Project administration:** James E. Haber.

**Resources:** Elena Sapède.

**Software:** Randall G. Tyers.

**Supervision:** Elena Sapède, James E. Haber.

**Writing – original draft:** Elena Sapède.

**Writing – review & editing:** Randall G. Tyers, Yuko Nakajima, James E. Haber.

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
