## [Decision Letter · Decision Letter 0]

8 Dec 2022

Dear Dr Haber,

Thank you very much for submitting your Research Article entitled 'Nonhomologous tails direct heteroduplex rejection and mismatch correction during single-strand annealing in Saccharomyces cerevisiae' to PLOS Genetics.

The manuscript was fully evaluated at the editorial level and by independent peer reviewers. The reviewers appreciated the attention to an important problem, but raised some substantial concerns about the current manuscript. Based on the reviews, we will not be able to accept this version of the manuscript, but we would be willing to review a much-revised version. We cannot, of course, promise publication at that time.

If you decide to revise the manuscript for further consideration at PLOS Genetics, please aim to resubmit within the next 60 days, unless it will take extra time to address the concerns of the reviewers, in which case we would appreciate an expected resubmission date by email to plosgenetics@plos.org.

We are sorry that we cannot be more positive about your manuscript at this stage. Please do not hesitate to contact us if you have any concerns or questions.

Yours sincerely,

Gregory P. Copenhaver

Editor-in-Chief

PLOS Genetics

Gregory Barsh

Editor-in-Chief

PLOS Genetics

Reviewer's Responses to Questions

**Comments to the Authors:**

Reviewer #1: Sapede et al. used an assay developed previously in the Haber lab to examine single-strand annealing (SSA) recombination between homologous or 3% divergent repeats separated by non-homologous sequence (tailed substrates). Previous work showed that SSA efficiency between divergent sequences was roughly 3-fold lower than between homologous sequences and that recombination between divergent sequences could be increased in mutants defective in mismatch recognition (msh6) or helicase activity (sgs1). Based on these observations Haber and colleagues proposed that the recognition of DNA mismatches in heteroduplex DNA formed during SSA initiated unwinding of the heteroduplex DNA, a mechanism termed rejection. Work also done by the Haber lab (Anand et al. Nature 2017, 544:377) showed that the presence of a 3’ non-homologous tail during break-induced recombination impacted heteroduplex rejection dependent on the mismatch recognition factor Msh2. In this system Msh2-dependent rejection required the presence of the non-homologous 3’ tail. The idea that a 3’ non-homologous tail could be important for rejection was raised earlier by Chakraborty et al. (Genetics 2016, 202:525) who provided evidence that 3’ non-homologous tail clipping during SSA is a critical regulatory step in the repair vs. rejection decision- rejection is favored before the 3’ tails are clipped.

Sapede et al. provide direct support for 3’ tails playing a critical role in heteroduplex rejection during genetic recombination. They constructed SSA substrates involving repeat sequences that are not separated by non-homologous sequence (labeled tailless substrates); thus 3’ non-homologous tails should not form during SSA. Recombination for the tailless substrates was induced through a Cas9-mediated double-strand break located precisely between the repeats.

The main conclusions were:

A. In contrast to tailed substrates containing divergent repeats, tailless substrates containing divergent repeats displayed low levels of heteroduplex rejection, though the weak rejection seen in tailless substrates was still dependent on MSH6 and SGS1 functions.

B. Curiously, SSA in the tailless construct showed dependence on MSH2, which was unexpected because a tail clipping step is not predicted to occur for this substrate.

C. The authors note an unusual position effect and bias for repair in the divergent constructs in which a DSB was initiated by either HO or Cas9 (Figure 4B, C). Interestingly, the position effect was lost in the divergent tailless strain, but a bias in correction was still seen.

D. The authors observed that SSA products in a msh2 null background displayed surprisingly high rates of mismatch correction, and opposite strand repair bias, a phenomenon which seems to be dependent upon the presence of non-homologous tails.

Comments

1. There is a lot of information presented in this paper; Table 1 alone required a serious effort. In general, the observations are quite interesting though at least for this reviewer there is a sense that the mechanism of mismatch correction in the different constructs has not been fully explored, though it would not be trivial to do so. Also, the authors wrote the paper as if the presence of a non-homologous tail impacting rejection is a new concept. The new work presented is not diminished by the previous studies. With this said, the authors should more clearly state in the introduction the previous work showing that the presence of tails during genetic recombination impacts rejection.

2. Throughout the paper, assays utilizing HO endonuclease and Cas9 endonuclease with a variety of guide sequences are compared to each another. This comparison is predicated on the near perfect efficiency and identical kinetics of cutting for each Cas9/guide RNA combination. Any less than perfectly efficient guides could inflate cell survival in this assay. The manuscript does not provide evidence to confirm that each guide behaves in the same manner. There is evidence to show in other systems that Cas9 cleavage efficiency varies widely between guide sequences (Doench et al. Nature Biotechnology 2016, 34:184, among many others), as well as a review of efficiency prediction algorithms for guide RNAs (Haeussler et al, Genome Biology 2016 17:148). To remedy this, the authors should analyze DSB formation upon galactose addition either by Southern Blot, comparing each unique guide RNA to each other and to the original HO endonuclease. I recognize that Cas9 expression is continuous when cells are grown in galactose, but there are examples where even under continuous selection for Cas9 expression cleavage is not efficient (e.g. Bao et al. ACS Synth Biol, 2015, 4:585). At the very minimum a Southern blot time course for the tailless substrate presented in Figure 2D should be done because the choice of a guide RNA in this situation is quite limited.

3. Figure 3B and Table 1: I am curious why the effect of a msh3 knockout wasn’t tested for the tailless divergent strain. Such an experiment would answer if the decreased viability seen for the msh2 knockout divergent tailless strain resulted from a defect in MSH2-MSH3 function (that could be explained in a model) or some other function that relied only on MSH2 (perhaps damage signaling?).

4. To better understand nature of the positional and biased mismatch repair effects shown in Figures 4A, B, S4, and S5, the authors could invert either the entire F-A construct in the chromosome or just the non-homologous DNA sequence. I recognize that a bias of repair was still seen in the tailless strain (Figure 4C), but this bias appeared weaker than seen for the tailed strain. The inversion experiments could really address what I feel is one of the most interesting aspects of the work.

Minor comments

1. The authors note that repair in the identical repeat tailed strains was lower compared to the identical repeat tailless strain. The decreased viability seen in the identical repeat tailed strains was seen previously in plating assays (for example, Goldfarb et al. Genetics 169:563) but not in Southern blot analysis (Sugawara PNAS 2004, 101:9315; Goldfarb Genetics 2005, 169:563). It’s probably worth indicating the disconnect between the genetic and physical assays, perhaps when the authors discuss the possibility of daughter cell survival loss.

2. Page 4, line 11. It is worth briefly mentioning that MutL homolog proteins have been implicated in heteroduplex rejection in other systems (Rayssiguier, Nature 1989, 342:396; Hum NAR 2019, 47:4554). The Hum and Jinks-Robertson study is particularly interesting; their experiments were performed in wild-type, msh6 (mismatch recognition) and mlh1 (mismatch processing) strains containing homologous and divergent substrates that allowed them to distinguish between crossover and non-crossover events. Their major observation was that mismatch recognition and processing could impact different kinds of recombination events involving divergent DNA sequences.

3. Methods section, “Analysis of SSA” and “Galactose-inducible Cas9 Endonuclease pRT02”

In the first section “Analysis of SSA” the DSB induction process is described as “HO endonuclease…. was induced by addition of 2% galactose to create a single DSB between the 200-bp repeated segments [10]. After induction of GAL::HO nearly all survivors were SSA products and could thus be monitored by colony counting. Cells were plated for individual colonies on YEPD and on YEP-GAL plates to induce endonuclease.”

In the second section, “Galactose inducible Cas9 endonuclease pRT02”, DSB induction is described “to induce the Cas9 endonuclease activity, cells were grown for 3h in YP-lactate, and then ~500 cells were plated on YEPD and YEP-Gal plates. This second section could be clarified. Does this mean that the Cas9 induced DSBs and the HO induced DSBs were treated differently in the SSA survival assays? If so, could we see a control to show that this difference has not affected SSA efficiency?

4. Figure 2 Panel A. Please specify that the tailed reaction involved GAL-HO and the Tailless GAL-CAS9. It’s vaguely written in the legend.

5. Figure 4B. Please indicate above the graph that this was performed in FA tailed strains.

Reviewer #2: Major comments:

Overall, I am enthusiastic about the subject area of this manuscript, and the authors’ conclusions, if true, would be very interesting. Frankly there are a lot of observations, some intriguing mechanistic hints, but the current manuscript seems to be lacking a many key experiments that are necessary to understand what is actually going on.

1. The key problem is the fact that the tailless construct is cut using Cas9, whereas the tailed construct is (most commonly) cut using HO. This makes it impossible to determine whether the differences that the authors are ascribing to the nonhomologous tail might actually be due to differences in the kinetics of Cas9 expression, DNA binding, and/or cleavage. Given that the role of the non-homologous tail in heteroduplex reject might entirely be due to timing, it’s crucial that the same nuclease be used for both tailed and tailless constructs. Unfortunately, this means the authors need to repeat many of their experiments with Cas9 cleavage. For some experiments (e.g. Fig 4) this has been done. For others it is simple (e.g. simply cut the tailed AA strain with Cas9 in Fig 2A). But for other experiments, this will require substantially more work (e.g. Figs 3,5,6).

2. Another unresolved problem with this manuscript is the fact that the repair biases are not understood. Repair in the Tailed SSA construct is biased towards the F repeat; when the repeats are swapped (Fig S5) repair becomes biased towards the A repeat. The authors conclude that “The bias to correct sequences in favor of F is in fact a bias in favor of whichever allele is located close to the left end of the annealed structure.” (pg. 14). But “close to the left end” really doesn’t really explain anything. Is this the asymmetry of the cut relative to the repeats? Does resection of the region between F and A expose strand-specific secondary structures? Is the bias due to something much more prosaic—for example the perfect homology at the 3’ end of the repeats is far longer than the perfect homology at the 5’ end? (The 5’ end has clustered sequence variants, see Fig S4 but a better diagram of this in Fig 1 is probably needed.)

3. The authors claim that differences in survival in Fig 2D between the various tailed intermediates argues that the presence of even a single flap stimulates heteroduplex rejection (pg 11). This is one of the major arguments of the paper. However, the authors cannot argue this solely with data from F-A strains. To prove that this survival effect is due to heteroduplex rejection, they must repeat the experiment in A-A strains and show that the effect of the flaps is restricted to strains where heteroduplexes can form.

4. The authors speculate about the role of the DNA polymerase delta proofreading activity on the correction of mispairs in this study (see for instance Fig 6, Fig S7). This really seems at odds with the effect of MMR defects (see Fig 5A, for example) where this sort of repair ought to be the _only_ mechanism for repair, yet there is no bias for the correction of MM6 or MM7 in msh6 or pms1 strains and unrepaired heteroduplexes predominate. These data suggest that the role of the proofreading activity is much more subtle and should actually be tested using appropriate pol3 point mutations.

5. The model the authors propose for mismatch correction in the heteroduplex is really intriguing. They argue (pg 19) that removal of the non-homologous flap creates a bias for the repair of that strand analogously to the use of strand discontinuities in MMR. Unfortunately, it appears that their own data are not consistent with the model. In supplemental figure 7, asymmetric cleavage to generate a one-flapped intermediate does not bias the repair to the flapless strand as this model would predict. A slightly modified model that would argue for blocking of the unflapped 3’ end with DNA polymerase might suggest that the flapless strand should be treated as the continuous strand and the Rad1-Rad10-cleaved flapped strand would then be the one repaired. In either case, the data in supplemental figure 7 doesn’t show the sort of symmetric change in bias that this model predicts.

Other comments:

1. Some changes to Figure 1 (and related figures) will probably improve the comprehension of the assays.

A. The authors should include the MATa site between the F and A repeats in Fig 1A like they do in Fig 2B so that it is clear that nothing has changed about the strain.

B. The authors need to indicate the distance between the cut site and the F and A repeats (particularly due to the biased repair in the tailed assay). Based on Fig. 2B, it appears that F is closer to the DSB than A.

C. The authors are missing an opportunity to make the Figure 1 (and later figures) really clear when they use different colored “X” characters to indicate the sequence variants between the F and A repeat. Why not use “F” instead of “X” for the F repeat and “A” instead of “X” for the A repeat? These make the diagrams match the outcomes in Fig 6 (for example) and make the figures understandable when printed in black & white.

D. The authors depict several SSA outcomes. In their “mixed repair” product “a”, they still have a heteroduplex that is not pointed out as in the unrepaired product “d”. The difference between “a” and “d” is that some sites in “a” are repaired, whereas none are in “d”. This should be clarified.

E. The authors probably should also drop the word “sectored” from their repair product “d”. Although it is true that the colonies are heterogeneous and are made up of distinct sectors after the original heteroduplex-containing strain divides, the word “sectored” implies use of a colony color sectoring assay which isn’t being used here and could be confusing. Perhaps “genetically heterogeneous colony” is what is really meant here (and is also relevant to “a”).

F. There is a fifth SSA outcome that they authors do not depict here. In Fig. 6 it’s clear that there are colonies with repair in which no heteroduplex is retained (e.g. the MM1,2,3 repaired as F and MM4,5,6,7 repaired as A). This outcome should be added.

2. The authors should compare their distribution of SSA outcomes from the NGS experiment in Fig. 6 with the outcomes derived from Sanger sequencing. Do these results give similar percentages at each site for lack of repair/repair to G/repair to A?

3. The way the genotypes are displayed in Fig 6A are really inconsistent. In the cases where initial heteroduplexes were observed, they show both strands. In cases without heteroduplexes, they only show one strand. The author should show both strands in all cases for consistency.

4. The authors should drop the analysis of MM2 throughout their data set. It’s clear that they don’t trust it (rightfully) due to the high frequency of deletion formation in long homopolymer runs in PCR (see their methods for “correcting” the data). The discussion regarding this variant (and Fig S8,9 and Fig 6 BC, which I believe must be from “uncorrected” sequencing data??) unnecessarily complicate the manuscript, as the status of MM2 does not play an important role for understanding the results.

5. The effects of the MutS homolog deletions in this assay are quite baffling and complicated by the different roles of Msh2-Msh3 and Msh2-Msh6 in the assay. Moreover, the fact that the msh2 deletion is very different than the msh3 msh6 double mutant is a really intriguing observation. The authors should really include the analysis of a msh2 msh3 double mutant (and possibly a msh2 msh3 msh6 triple mutant if the msh2 msh3 double mutant doesn’t look like the msh2 and msh2 msh6 mutants) for the tailed assay in Fig 5 and Fig S3 and increase the number observations of the msh3 msh6 and msh2 msh6 double mutants. The authors infer that the msh2 spectrum is a failure of both MSH3- and MSH6-dependent tail removal and mispair correction pathways, but given the difference between all of the mutants, it’s not clear why the msh2 spectrum should be different than the msh3 msh6 double mutant spectrum.

6. The authors state that “Evidence that GAL::CAS9 cleavage might be less efficient and possibly delayed…” (pg 14, bottom). There are no experiments showing delay of cleavage. The only argument that they appear to be making is that the presence of both alleles suggests “a delay in Cas9 cutting” (Fig S6 legend). This argument simply makes no sense. Whether or not repair of mispairs occur on the heteroduplex must, by definition, occur after the heteroduplex is formed (involving DSB formation and resection). Unless, of course, the author’s amplification/sequencing protocols cannot distinguish between unrepaired heteroduplexes involved in SSA and cells in which no rearrangements occur.

7. Are the colors of the repeats swapped in Fig. S7 or were these experiments performed on the A-F strain?

8. Figure S4 and the description of the nFA strain (pg. 13) don’t appear to match. In the text the authors claim that they “replaced the deletion of the 1T in the ‘A’ sequence by inserting a G at position 23”. From this description, I would expect that the insertion would eliminate the deletion in the ‘A’ sequence and shift the ‘A’ sequences over. However Figure S4 still shows that the modified sequence is the F sequence, not the A sequence (e.g. they “inserted a G at position 23 in the ‘F’ sequence”). Also, the nFA sequence appears to have a deletion of a 1T so that there is no sequence length difference with A.

Reviewer #3: This is a review of PGENETICS-D-22-01303, "Nonhomologous tails direct heteroduplex rejection and mismatch correction during single-strand annealing in Saccharomyces cerevisiae." This study examines genetic regulation of heteroduplex rejection during single strand annealing (SSA) / repeat mediated deletions. SSA is a DNA double strand break (DSB) repair outcome that involves repeats that flank the DSB that anneal to each other, causing a deletion between the repeats and loss of one copy of the repeats. SSA associated with human disease is unlikely to involve identical repeats, and as such how repeat divergence affects the mechanism of these events (i.e. heteroduplex rejection) is an important question in genome stability. The authors describe a straightforward genetic assay for SSA in yeast that enables examination of two key phenomenon 1) effect of DNA tail on heteroduplex rejection, 2) the pattern of heteroduplex/mismatch resolution. DNA tail is varied by placing the initiating chromosomal double-strand break at various distances from the homologous repeats. The pattern of heteroduplex rejection is examined with both pooled sequencing, as well as sequencing of individual clones that enables analysis of sectored colonies. Several interesting findings are shown: 1) heteroduplex rejection (and hence the influence of mismatch repair / MSH6, as well as the SGS1 helicase) requires a DNA tail, 2) for the events that show partial mismatch correction, there is a gradient that "favoring the sequence opposite the 3’ end of the annealed strand," which again is dependent on a DNA tail. Altogether this study provides novel insight into heteroduplex rejection, along with mechanisms of mismatch correction during DNA repair. The results are clearly presented, with limitations of the approach clearly described (e.g. the limitations inherent in the +T mismatch #2). The Discussion is clear and will stimulate research in this area.

The main concern is the presentation of Figure 6 is relatively hard to follow. Two recommendations:

1. To the novice reader, I think the figure legend and "description" of the SSA genotypes may be difficult to understand. I recommend creating a supplemental figure with illustrations for each genotype that shows the heteroduplex, the likely repair outcome, and then the daughter cells. For example, showing the heteroduplex repaired to FFFFFF/FFFFFF then leading to identical daughter cells, and hence not a mixed colony, would help the novice reader. A limitation of this approach is that such illustrations are models that of course might be wrong, but if clearly described as models, I think this clarity will help readers substantially understand the data.

2. There is a lot of data on a small graph on 6A. Perhaps keep Figure 6A as a whole figure and lengthen it to make the bars thicker, and move 6B/6C to supplemental. Alternatively / in addition, split MSH2 and MSH6 into separate graphs.

One minor concern:

3. The cartoon with the 7 X's is not to scale, since the mismatches are not equidistant. I recommend a simple figure under Fig 1B that shows a larger version of the 7 X's illustration but the position of the X's roughly to scale. Of course, the positions of the mismatches are in the table in Fig 1B, put it would be great to have this in an illustration to get a visual of the structure, and if the authors use the same color/X's schematic as the cartoon used throughout the study, it can help reinforce the relative position of the mismatches in the model figure.

**Have all data underlying the figures and results presented in the manuscript been provided?**

Reviewer #1: Yes

Reviewer #2: Yes

Reviewer #3: Yes

PLOS authors have the option to publish the peer review history of their article (what does this mean?). If published, this will include your full peer review and any attached files.

Reviewer #1: No

Reviewer #2: No

Reviewer #3: No

---

## [Decision Letter · Decision Letter 1]

28 Dec 2023

Dear Jim,

We are pleased to inform you that your manuscript entitled "Nonhomologous tails direct heteroduplex rejection and mismatch correction during single-strand annealing in Saccharomyces cerevisiae" has been editorially accepted for publication in PLOS Genetics. Congratulations!

Please see Reviewer #1's comments below which describe a couple of very minor issues that you should attend to as you prepare your final draft for the production team (the editorial team will not need to re-evaluate).

Yours sincerely,

Gregory P. Copenhaver, Ph.D.

Editor-in-Chief

PLOS Genetics

Gregory Barsh

Editor-in-Chief

PLOS Genetics

Comments from the reviewers (if applicable):

Reviewer's Responses to Questions

**Comments to the Authors:**

Reviewer #1: The authors have done a nice job responding to my concerns. They addressed my major concerns #1 and 2 and the new work regarding concern #3 further revealed a role for Msh2 in the absence of its partners that can be explored in the future. While it would have been nice to see an inversion experiment, the authors provide additional explanations for their observations that make sense. Also, the expanded work on the phenotype of rad52 mutants in the tailless constructs are a very nice addition.

Some minor comments.

1. In the abstract: Remarkably, heteroduplex rejection is very low in strains where the identical repeats were immediately adjacent (Tailless strains)…..” I believe that the authors meant to write “divergent repeats”, not “identical repeats.”

2. Page 11 top of the page. I believe that the authors are referring to Figure 2D but this panel is not cited in the paragraph. Also, Figure 2B is not cited in the manuscript. As an aside, Figure 2A is superfluous (data are in Figure 2C) and can be removed.

Reviewer #3: The authors have carefully revised the study to address certain control issues raised by the other reviewers, and clarification issues raised by all reviewers. Additionally, novel RAD52-dependence data has been added, which increases the impact of an already impactful study. The substantial influence of non-homologous tails on HR/SSA fidelity/mechanism remains a relatively underappreciated aspect of genome stability, and I predict this study will be a landmark advance that stimulates further research in this area.

**Have all data underlying the figures and results presented in the manuscript been provided?**

Reviewer #1: Yes

Reviewer #3: Yes

PLOS authors have the option to publish the peer review history of their article (what does this mean?). If published, this will include your full peer review and any attached files.

Reviewer #1: No

Reviewer #3: No

**Data Deposition**

http://datadryad.org/submit?journalID=pgenetics&manu=PGENETICS-D-22-01303R1

**Press Queries**

---

## [Editor Report · Acceptance letter]

30 Jan 2024

PGENETICS-D-22-01303R1 

Nonhomologous tails direct heteroduplex rejection and mismatch correction during single-strand annealing in Saccharomyces cerevisiae 

Dear Dr Haber, 

We are pleased to inform you that your manuscript entitled "Nonhomologous tails direct heteroduplex rejection and mismatch correction during single-strand annealing in Saccharomyces cerevisiae" has been formally accepted for publication in PLOS Genetics! Your manuscript is now with our production department and you will be notified of the publication date in due course.

With kind regards,

Bernadett Koltai

PLOS Genetics

On behalf of:
